# Epithelial-mesenchymal plasticity determines estrogen receptor positive breast cancer dormancy and epithelial reconversion drives recurrence

Patrick Aouad [1], Yueyun Zhang[1], Fabio De Martino[1], Céline Stibolt[1], Simak Ali [2], Giovanna Ambrosini[1], Sendurai A. Mani[3], Kelly Maggs[4], Hazel M. Quinn [1], George Sflomos [1] & Cathrin Brisken [1,5] ✉

More than 70% of human breast cancers (BCs) are estrogen receptor α-positive (ER⁺). A clinical challenge of ER⁺ BC is that they can recur decades after initial treatments. Mechanisms governing latent disease remain elusive due to lack of adequate in vivo models. We compare intraductal xenografts of ER⁺ and triple-negative (TN) BC cells and demonstrate that disseminated TNBC cells proliferate similarly as TNBC cells at the primary site whereas disseminated ER⁺ BC cells proliferate slower, they decrease *CDH1* and increase *ZEB1,2* expressions, and exhibit characteristics of epithelial-mesenchymal plasticity (EMP) and dormancy. Forced E-cadherin expression overcomes ER⁺ BC dormancy. Cytokine signalings are enriched in more active *versus* inactive disseminated tumour cells, suggesting microenvironmental triggers for awakening. We conclude that intraductal xenografts model ER + BC dormancy and reveal that EMP is essential for the generation of a dormant cell state and that targeting exit from EMP has therapeutic potential.

Breast cancer (BC) is the most commonly diagnosed malignancy worldwide[1]. Clinical management of BC relies on grade, stage and tumor subtype, which can be defined by histology and immunohistochemistry (IHC) as well as by gene expression profiling. More than 70% of BC cases are estrogen receptor α-positive (ER⁺). ER⁺ BCs tend to be of lower grade and lower proliferative indices than ER⁻, including HER2⁺ and ER⁻PR⁻HER2⁻ (triple-negative, TN), BCs. Accordingly, patients with ER⁺ BC have better 5-year survival rates than patients with ER⁻ BCs. However, while ER⁻ BC patients who do not relapse within the first five years after treatments are generally considered disease-free, ER⁺ BC patients remain lifelong at risk for relaps despite benefiting from endocrine therapy initially[2].

Disseminated tumor cells (DTCs) that leave primary tumors early during tumorigenesis and remain dormant at distant sites contribute to delayed recurrence[3]. The existence of dormant DTCs is supported by clinical observations that recipients of organs transplanted from individuals with undiagnosed malignancies developed the corresponding types of tumor[4,5]. It is critical to understand the mechanisms governing dormancy for preventative intervention; however, the issue is understudied due to the lack of suitable in vivo models[6–9]. Most studies of the mechanisms underlying metastasis in vivo used genetically engineered mouse models (GEMMs) or xenograft models. Regarding mammary tumorigenesis, most GEMMs develop ER⁻ tumors[10–12] and most xenograft studies have relied on subcutaneous

[1]ISREC - Swiss Institute for Experimental Cancer Research, School of Life Sciences, Ecole Polytechnique Fédérale de Lausanne (EPFL), CH-1015 Lausanne, Switzerland. [2]Division of Cancer, Department of Surgery & Cancer, Imperial College London, Hammersmith Hospital Campus, London, United Kingdom. [3]Department of Translational Molecular Pathology, MD Anderson Cancer Center, Houston, TX 77030, USA. [4]Laboratory for Topology and Neuroscience, Brain Mind Institute, EPFL, CH-1015 Lausanne, Switzerland. [5]The Breast Cancer Now Toby Robins Breast Cancer Research Centre, The Institute of Cancer Research, London, UK. ✉e-mail: cathrin.brisken@epfl.ch

grafting or injecting BC cells directly into the bloodstream or at distant sites but these approaches poorly reflect the clinical situation.

We recently reported that grafting human ER[+] BC cell lines and patient-derived xenografts (PDXs) into the milk ducts of immune-compromised mice (MIND) substantially improves take rates over the traditional subcutaneous engraftment[13,14]. MIND models recapitulate human ER[+] BC progression, from in situ stage to spontaneous dissemination to clinically relevant distant organs[13,14] and reflect specific histopathological subtypes, including the invasive lobular carcinoma[15,16]. Bioluminescence emanating from ER[+] DTCs can be detected in different organs when tumors still appear to be confined within the milk ducts, suggesting early dissemination[13,14]. While DTC load increases with prolonged tumor growth, no macro-metastasis is detected up to seven months after engrafting MCF-7 cells[14] and >1 year after engrafting ER[+] PDXs[13].

Here, we compare ER[+] and TN BC progression in vivo using MIND models and show that dormancy is specific for ER[+] DTCs and is driven by Epithelial-Mesenchymal Plasticity (EMP). Forced exit from EMP can overcome dormancy, providing a potential target against latent disease.

## Results

### Tumor growth and progression in ER[+] *versus* TN MIND models

In ER[+] BC MIND models, DTCs detected in clinically relevant distant organs failed to progress to macro-metastases[13,14,17]. To determine whether this observation reflects ER[+] BCs tumor dormancy or is an experimental artifact, we compared the metastatic behavior of ER[+] and ER[−] BC cells by the MIND approach. We modeled ER[−] BC using TNBC cell lines BT20 and HCC1806 (Table 1) as well as a PDX derived from an untreated primary TNBC, T70 (Table 2). To model ER[+] BC, we used the established ER[+] cell lines MCF-7 and T47D (Tables 1) and 2 ER[+]PR[+]HER2[−] PDXs: T99 derived from an untreated primary tumor and METS15 derived from the ascites of a patient with advanced disease (Table 2). All tumor cells were infected with lentiviruses expressing *GFP-Luc2* or *RFP-Luc2* and injected into the milk ducts of NOD.Cg-*Prkdc*[scid] *Il2rg*[tm1Wjl]/SzJ (*NSG*) females (Fig. 1a)[14]. Take rates, defined as the percentage of glands showing in vivo tumor cell growth relative to the total number of glands injected, were 100% for TNBC and ≥90% for ER[+] BC xenografts (Fig. 1b). Within five weeks, mice xenografted with BT20, HCC1806, and T70 developed palpable mammary tumors, increased bioluminescence 702, 592, and 1915 folds (Fig. 1c, d), they showed signs of morbidity and were euthanized. Over the same time period, bioluminescence from MCF-7, T47D, T99, and METS15 xenografts increased only 42, 39, 54, and 5 folds, respectively (Fig. 1c, d). Five months later, bioluminescence increased 1000 to 2000 folds and tumors became palpable but host mice remained healthy (Fig. 1c, e).

Ex vivo imaging of resected organs from ER[+] and TN BC models revealed bioluminescence in the brain, lungs, liver, and bones; signals from BT20 and HCC1806 cells in the lungs were particularly high (Fig. 1f, g). T70 cells derived from a patient with pregnancy-associated BC were readily detected in the liver (Fig. 1f), in line with clinical reports of increased liver metastasis in these patients[18]. Bioluminescence

signals in the lungs and bones resected from mice bearing MCF-7, T47D, or MET15 cells were high, whereas those signals from T99 cells were just above background levels in 3 out of 10 mice (Fig. 1g), consistent with previous observations[13]. At the respective experimental endpoints, primary tumor burdens were comparable, but bioluminescence of the lungs relative to that of the mammary glands was on average 100-fold higher in TNBC models than in ER[+] BC models (Fig. 1h). Thus, metastases in mice with TNBC xenografts progressed faster than in mice with ER[+] grafts, reflecting the clinically observed subtype difference in BC metastasis.

We then examined early tumor cell seeding by fluorescence stereo microscopy of mammary glands xenografted with BT20 and HCC1806 cells. One week after intraductal injection, the GFP signal was confined to the ducts (Supplementary Fig. 1a, b) and histological examination confirmed that tumor cells were within the ducts as is characteristic of the in situ stage of the human disease (Supplementary Fig. 1c, d). No disruption of the epithelial-stromal border was detected but the fibrous extracellular matrix (ECM) around ducts filled with tumor cells was thicker than around those without human cells (Supplementary Fig. 1c, d). At this stage, ex vivo bioluminescence of the lungs was above background levels in 5 of 10 BT20- and 2 of 4 HCC1806-bearing mice (Supplementary Fig. 1e). Two weeks after engraftment, multiple invasive foci were detected in the engrafted mammary glands (Supplementary Fig. 1a–d). In all BT20- and HCC1806-bearing mice, the lungs showed increased bioluminescence (Supplementary Fig. 1e) and GFP[+] lesions were detected by fluorescence stereomicroscopy (Supplementary Fig. 1f). Thus, both TN and ER[+] BC intraductal xenografts disseminate during the in situ stage as observed in clinical studies[19,20] but only TN DTCs progress to macro-metastases.

### ER[+] metastatic lesions are dormant

The slow increase in bioluminescence at the distant sites in the ER[+] BC models suggested dormancy. Hence, we analyzed DTCs in the lungs by fluorescence stereo microscopy and subsequent histological analysis. Both BT20 and HCC1806 cells gave rise to lesions that were readily detected by fluorescence stereo microscopy and measured 0.5-3.0 mm in diameter on histological examination (Fig. 2a, b). In contrast, lung lesions arising from MCF-7 and T47D intraductal xenografts were barely detectable by fluorescence stereo microscopy and merely 10–100 μm in diameter (Fig. 2a, c). In the case of the TN PDXs, liver lesions were 0.5–1.0 mm in diameter (Fig. 2d). METS15 cells were detected in the lung by both approaches (Fig. 2e, f), but T99 cells were neither detected by fluorescence stereo microscopy nor by the subsequent screening of >40 histological sections from lung lobes of 3 host mice, in agreement with the low bioluminescence (Fig. 1d).

Next, we quantified the proliferative indices of primary and metastatic lesions (Supplementary Table 1). The Ki67 indices for BT20 and HCC1806 lung lesions, independent of their sizes, were comparable to those of primary tumors (Fig. 2g, h, Supplementary Fig. 2a). In contrast, the Ki67[+] indices in ER[+] distant lesions were on average only one-third of those in primary tumors (Fig. 2i and Supplementary Fig. 2b).

To characterize cell cycle dynamics, we used a fluorescent ubiquitination-based cell cycle indicator (FUCCI)[21], which distinguishes non-cycling ($G_0/G_1$) cells as red fluorescent from cycling (S-$G_2$-M) cells as GFP[+] or double-positive cells (Fig. 2j). Scoring of red versus green and yellow fluorescent cells in xenografted MCF-7:FUCCI-bearing mice showed that 38.6% of cells in the primary tumors were in $G_0/G_1$ phase and this increased to 68.2% ($P < 0.05$) in matched lung lesions (Fig. 2k, l).

To test for dormancy, we co-immunofluorescence labeled lungs for p27, Ki67, and CK8. p27 is a marker of dormancy[22] whereas CK8 was used to distinguish DTCs (CK8[+]) from the surrounding mouse alveolar cells (CK8[LOW]). Less than 1% of CK8[+] TNBC cells were p27[+]Ki67[−] whereas more than 10% of ER[+] BC cells were p27+Ki67-

## Table 1 | Cell lines used in this study

| Cell line | Age (years) | Source | ER | PR | HER2 | Number of passages after obtention from ATCC |
|---|---|---|---|---|---|---|
| MCF-7 | 69 | Pleural Effusion | + | + | − | 7–30 |
| T47D | 54 | Pleural Effusion | + | +, Amp | − | 6–8 |
| BT20 | 74 | Primary Tumor | − | − | − | 6–11 |
| HCC1806 | 60 | Primary Tumor | − | − | − | 20–23 |

*Amp* amplification, + positive, − negative.

**Table 2 | PDXs and corresponding patient tumor characteristics**

| PDX | Age at Surgery | Tumor Type | ER% | PR% | HER2% | Ki67% | Treatment | Generation in Mice |
|---|---|---|---|---|---|---|---|---|
| T70 | 39 | NST, Primary | negative | negative | negative | >90 | untreated | 6 to 8 |
| T99 | 57 | NST, Primary | 57 | 70 | negative | 20 | untreated | 6 to 8 |
| METS15 | 59 | NST, Ascites | 90 | 100 | negative | N/A | chemo, AI, fulvestrant | 2 to 4 |

*NST* no special type, *N/A* not applicable, *chemo* chemotherapy, *AI* aromatase inhibitors.

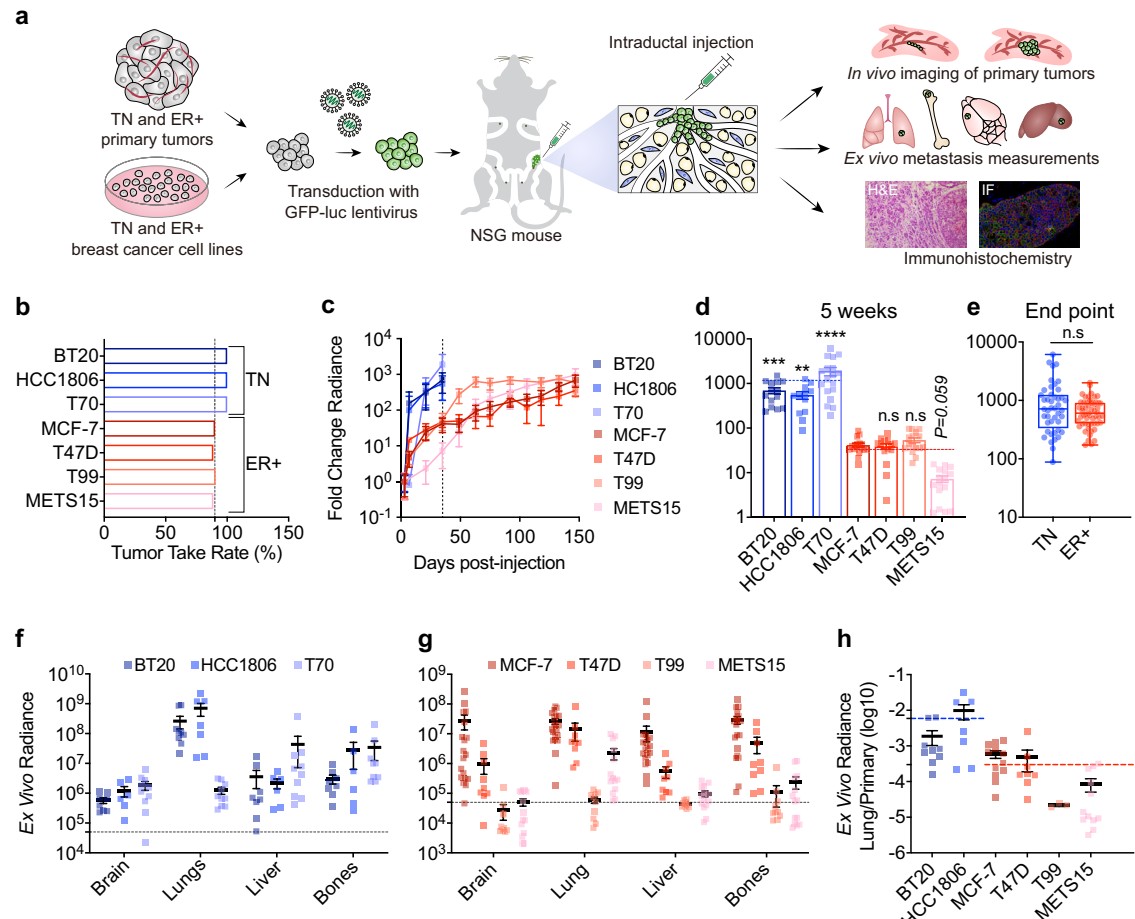

**Fig. 1 | ER+ and TN BC cells show distinct growth and metastatic behavior.**
**a** Scheme illustrating the intraductal xenografting approach used in this study.
H&E: Haematoxylin & Eosin, IF: Immunofluorescence. **b** Bar graph showing take
rates for TN (blue) and ER+ (red) BC cells injected intraductally, 13–19 mammary
glands of 5–9 mice injected in each group. The vertical dashed line indicates 90%.
**c** Graph showing the fold-change of bioluminescence over time for all intraductal
xenografts. Data represent mean ± SEM of 13-19 mammary glands from 5–9 mice in
each group. The dashed line indicates the experimental end-point for TNBC
xenografts. **d** Fold-change of bioluminescence at 5 weeks after intraductal injec-
tion. Data represent mean ± SEM of 13–19 mammary glands from 5 to 9 mice in
each group. Blue and red dashed lines represent the average change biolumines-
cence of TN (1148 fold) and ER+ BC (32 fold), respectively. One-way ANOVA,

Kruskal-Wallis test relative to MCF-7 (control). **e** Box plot showing the fold-change
of bioluminescence at endpoint for TN (5 weeks) and ER+ (5-6 months) BC cells
from Fig. 1c. Boxes span the 25th to 75th percentile, whiskers 1.5 times the inter-
quartile range. Boxplot whiskers show minimum and maximum values. Two-tailed
Mann-Whitney test. **f**, **g** Bar plot showing ex vivo bioluminescence of resected
organs from 9, 7, and 6 mice bearing BT20, HCC1806, and T70 xenografts,
respectively, **f** and 20, 9, 10, and 14 mice bearing MCF-7, T47D, T99, METS15
xenografts, respectively, **g** Data represent mean ± SEM. **h** Dot plot showing the
ratio of bioluminescence in lungs over primary tumor. Data represent mean ± SEM
of n = 9 (BT20), 7 (HCC1806), 12 (MCF-7), 8 (T47D), 3 (T99) and 12 (METS15) mice.
Blue and red dashed lines represent the average ratio for TN and ER + BC cells,
respectively.

(Fig. 2m, n and Supplementary Fig. 3a, b). Thus, a substantial fraction
of ER+ DTCs are arrested in the $G_0/G_1$ phase of the cell cycle as
characteristic of a quiescent and/or dormant state.

**DTCs have features of epithelial-mesenchymal plasticity (EMP)**
Individual ER+ lung DTCs appeared mesenchymal (Fig. 3a), prompting
us to quantify the cellular aspect ratio (CAR), as major axis: minor
axis[23], of tumor cells. A CAR > 1.7 characteristic of mesenchymal cells
characterized 10-15% of tumor cells at the primary site Fig. 3b and

40–50% of the DTCs in the lungs (Fig. 3c). Next, we analyzed the E-cad
expression by IF. The E-cad protein was readily detected in primary ER+
BC cells but signal intensity was reduced to <10% in matched lung
DTCs, which were identified based on CK8 expression (Fig. 3d and
Supplementary Fig. 4a–c). We micro-dissected fluorescent lesions
from distant sites and compared their mRNA levels to those of the
respective primary tumors by real-time PCR (RT-PCR) analysis using
human-specific primers. *MKI67* and *CDH1* transcript levels were sig-
nificantly lower in distant lesions than in primary tumors (Fig. 3e, f,

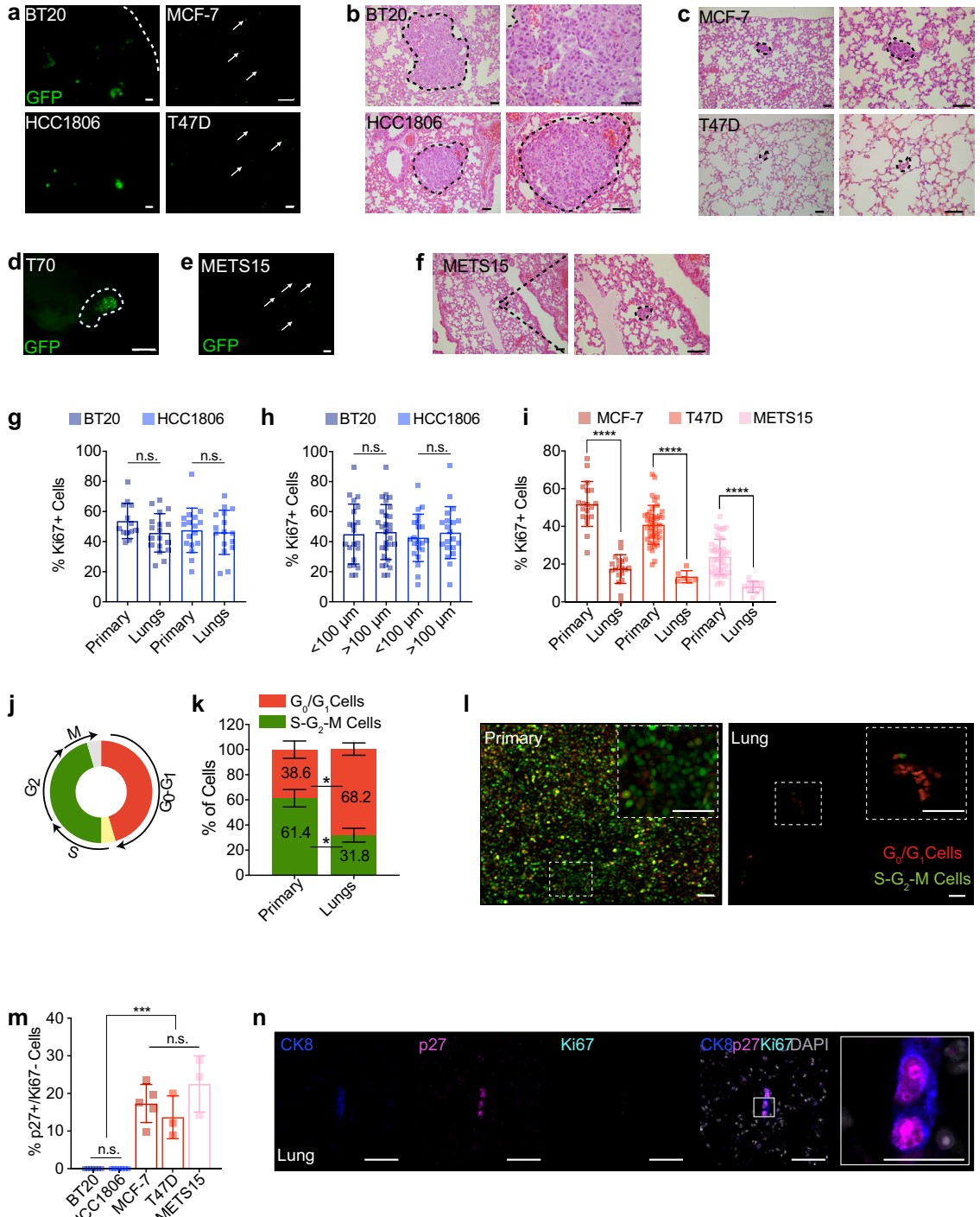

**Fig. 2 | ER⁺ metastatic lesions are dormant. a** Representative fluorescence stereo micrographs of lungs from ≥ 3 mice with BT20, HCC1806, MCF-7, or T47D intraductal xenografts, arrows point to DTCs. Scale bar, 1 mm. **b, c** Representative micrographs of H&E stained lung sections from 3 mice bearing BT20 and HCC1806 (**b**) or MCF-7 and T47D (**c**) intraductal xenografts. Scale bar, 50 μm. **d, e** Representative fluorescence stereo micrographs of the liver (**d**) and lungs (**e**) from ≥3 mice bearing T70 and METS15 MIND xenografts. Scale bars, 1 mm. **f** Representative micrographs of H&E stained lung sections from 3 mice bearing METS15 intraductal xenografts. Scale bars, 50 μm. **g** Percentage of Ki67⁺ cells in matched primary and lung sections from mice 5 weeks after intraductal injection of BT20 and HCC1806 cells, $n ≥ 14$ sections. **h** Bar graph showing Ki67 index of BT20 and HCC1806 lung lesions of different sizes. **i** Ki67⁺ index in matched primary tumor and lung sections from mice 5 months after intraductal injection of MCF-7, T47D, and METS15 cells as indicated. In **g** and **i** each data point represents ≥ 1,000 and 100 cells analyzed,

mean ± SD from ≥3 mice/condition, respectively. **g–h** Data represent mean ± SD from 3 different hosts. Student's unpaired t-test, two-tailed. **j** Scheme of FUCCI reporter. **k** Bar plot showing percentage of cycling and non-cycling cells in primary tumors and in lung micro-metastases in MCF-7 intraductal xenografts-bearing mice. Data represent mean ± SD from 3 host mice. Paired t-test. **l** Representative fluorescence micrographs of MCF-7:FUCCI cells in matched primary tumor (left) and lung (right). Scale bars, 50 μm. **m** Bar plot showing the percentage of p27⁺Ki67⁻ cells over total human cells in the lung. Data represent mean ± SD from $n ≥ 3$ mice, and dots represent ≥40 cells analyzed. One-way ANOVA. **n** Representative immunofluorescence micrographs for CK8 (blue), p27 (magenta) and Ki67 (cyan), counterstained with DAPI (gray) on lung section from MCF-7-bearing mice. Scale bar, 50 μm; inlet, 20 μm. *, ***, ****, and n.s represent $P < 0.05$, 0.001, 0.0001, and not significant, respectively.

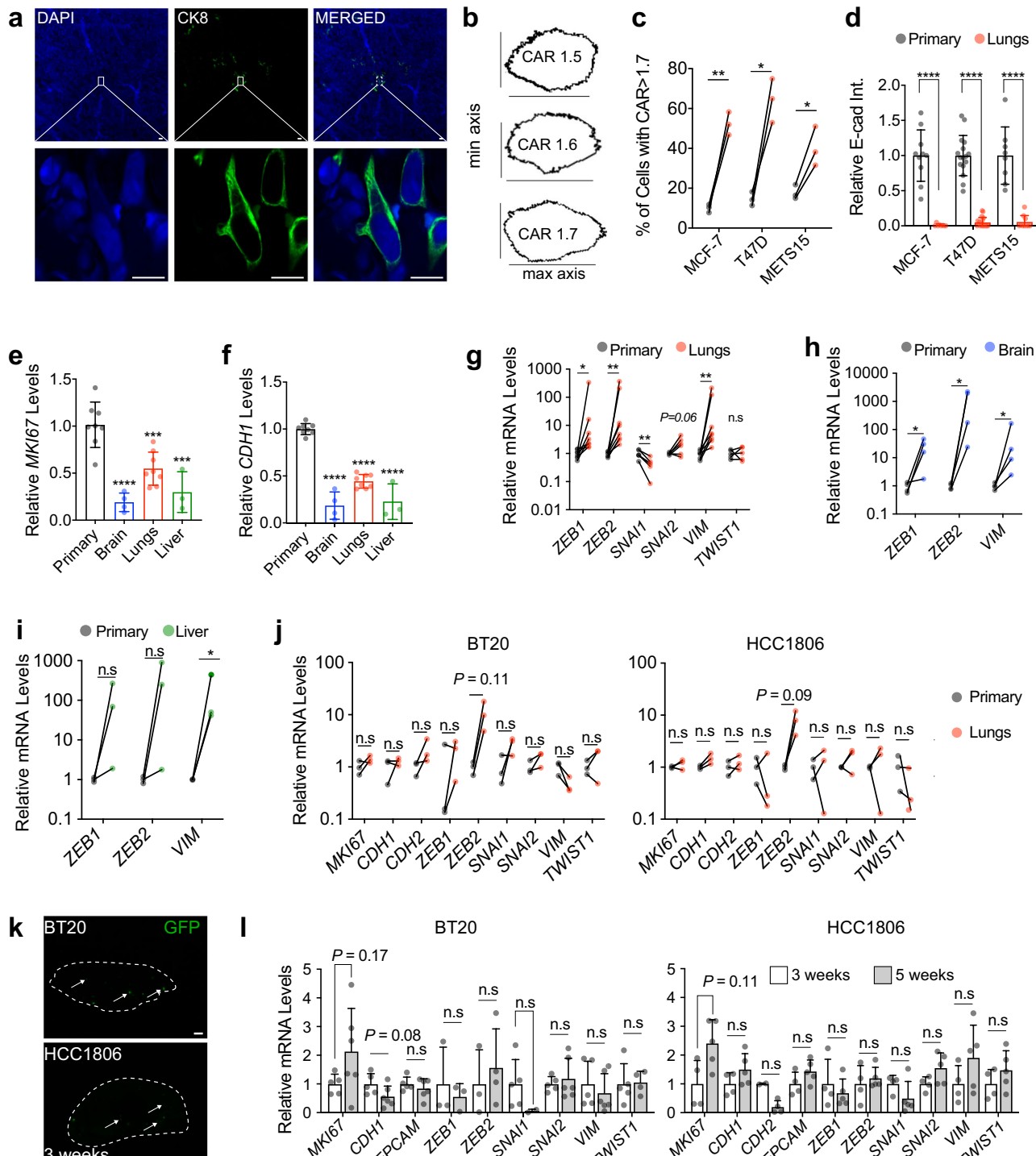

**Fig. 3 | Dormant DTCs from ER⁺ intraductal xenografts have an EMP signature.**
**a** CK8 staining on optically-cleared sections of METS15 cells in host's lungs. Scale bars, 10 μm. **b** Representative masks representing different cellular aspect-ratio (CAR) with respect to the morphology **c** Percentage of cells with CAR > 1.7 in matched primary tumors and in the lung from intraductal ER⁺ BC xenografts-bearing mice. Data represent mean ± SD, each data point represents at least 100 primary cells and 10 lung DTCs from 3 mice. Paired Student's *t*-test. **d** Relative E-cad intensity (Int.) in matched primary tumor cells and lung DTCs in mice bearing indicated intraductal xenografts. Data represent n≥8 images, mean ± SD from 3 mice. Students t-test **e, f** Relative *MKI67* (**e**) and *CDH1* (**f**) mRNA levels in MCF-7 cells in the primary tumor (8 mice) or in the lung (*n* = 8), brain (*n* = 4), and liver (*n* = 3) DTCs. Data represent mean ± SD. One-way ANOVA. **g–i** Relative levels of indicated

mRNAs in matched MCF-7 primary tumors and lung (**g** *n* = 8), brain (**h** *n* = 4), and liver (**i** *n* = 3) DTCs. Wilcoxon test. **j** Relative levels of indicated mRNAs in matched BT20 or HCC1806 primary tumors and lung metastases from 3 mice. Paired t-test. **k** Representative fluorescence stereo micrographs of lungs from at least 3 mice 3 weeks after intraductal injection of BT20 or HCC1806 cells. Arrows point to micro-metastases. Scale bar, 1 mm. **l**. Relative mRNA levels of the selected genes in 3 *versus* 5-6 weeks lung DTCs retrieved from at least 4 mice bearing BT20 (left) or HCC1806 (right), *n* ≥ 4. Data represent mean ± SD, Wilcoxon test. Gene expression was normalized to the geometric mean of *GAPDH* and *HPRT* in panels **e–j**, and **l**. *, **, ***, ****, and n.s represent *P* < 0.05, 0.01, 0.001, 0.0001, and not significant, respectively.

Supplementary Fig. 4d). Transcript levels of the EMT transcription factors (EMT-TFs), *ZEB1* and *ZEB2*, and the mesenchymal intermediate filament *VIM* were significantly higher in lung lesions than in primary tumors (Fig. 3g, Supplementary Fig. 4d). *SNAI2* and *TWIST1* mRNA levels did not differ between primary tumors and metastatic lesions whereas *SNAI1* expression was down-modulated in metastatic lesions in the MCF-7 model (Fig. 3g). *CDH2, SNAI1, SNAI2*, and *TWIST1* transcripts were not detected in either primary tumors or lung lesions in the METS15 model. *ZEB1, ZEB2*, and *VIM* transcript levels were also upregulated in brain DTCs (Fig. 3h). In the 3 liver lesions that we isolated, *VIM* transcripts were significantly up-regulated (*P* < 0.05*), ZEB1 and ZEB2* transcript levels showed a trend to increase (*P* < 0.12 and *P* < 0.25, respectively) (Fig. 3i). As an alternative approach to microdissection, we performed qRT-PCR using primary tumours and lung tissues from METS15 xenograft-bearing mice. Similar alterations in expression levels of human-specific genes in lung DTCs compared to their matched primary tumors were observed (Supplementary Fig. 4e).

In BT20 and HCC1806 models, *MKI67, CDH1, CDH2, ZEB1, SNAI1, SNAI2, VIM*, and *TWIST1* transcript levels were comparable between matched primary site and lungs with macro-metastases (Fig. 3j), suggesting that dormancy was not a feature of TNBC metastases. To determine whether TN DTCs pass through a dormant state prior to becoming macro-metastases, we compared expression levels of epithelial and EMP-marker genes in the lungs at 3 weeks, when micro-metastases prevailed (Fig. 3k), *versus* 5-6 weeks, when macro-metastases were detected, after intraductal injection with BT20 and HCC1806 cells. At both time points, similar expression levels of *MKI67, CDH1, EPCAM*, and EMT-TFs, and *VIM* were observed in lungs (Fig. 3l). In addition, the proliferative TNBC cells were in comparable EMP states in the primary (Supplementary Fig. 4f) and distant sites. Thus, mesenchymal morphology and expression of multiple EMP markers are features specific to ER⁺ DTCs.

## Role of EMP in ER⁺ tumor progression

Accumulating evidence supports the notion that tumor cells assume different EMP states that are important for their invasion and metastasis[24–27]. To assess whether EMP occured during the transition from in situ to invasive stage in ER⁺ BC, we measured *CDH1* and EMT-TFs expression 1 month after intraductal injection of MCF-7 cells, when primary tumors were in situ, and at 5 months, when they were invasive (Fig. 4a). *CDH1, ZEB1*, and *VIM* transcript levels did not change whereas the *ZEB2* expression decreased and *TWIST1* and *SNAI2* expressions increased at the invasive stage (Fig. 4b). IF showed that MCF-7 cells expressed E-cad and CK8 proteins in in situ and invasive lesions (Fig. 4c) as well as in areas of tumor budding, at the leading invasive edge, and in proximity to blood vessels (Supplementary Fig. 5a-c). Thus, EMP features characteristic of ER⁺ DTCs are not readily detected at the primary site in our models.

To test whether EMP favored the metastatic spread of ER⁺ BC cells early in tumor progression, we induced EMP by reducing *CDH1* expression or by overexpressing *ZEB1*. Given the heterogeneity of the lentivirally transduced polyclonal cell populations, we expected that a whole spectrum of EMP states would be produced. Control cells formed epithelial islands with cobblestone morphology whereas MCF-7:sh*CDH1* cells showed discohesive growth (Supplementary Fig. 5d, e), decreased cell proliferation (Supplementary Fig. 5f) and increased the *ZEB1* levels. Expression of *ESR1, PGR* or *AR* was unaltered (Supplementary Fig. 5g). *ZEB1* overexpression (Supplementary Fig. 5h) had similar effects on cell morphology and proliferation (Supplementary Fig. 5h–j), reduced *CDH1, ESR1*, and *PGR* expression, and increased *CDH2* expression (Supplementary Fig. 5k). Thus, both MCF-7:sh*CDH1* and MCF-7:*ZEB1* cells acquire distinct EMP features defined by recent guidelines[28].

When grafted intraductally, MCF-7:sh*CDH1* cells grew less than MCF-7:shSCR controls (Fig. 4d). Mammary gland weight and E-cad

expression at end-point were reduced (Supplementary Fig. 6a–c). Analysis of H&E stained sections showed that MCF-7:shSCR cells invaded the stroma whereas MCF-7:sh*CDH1* cells remained mostly in situ (Fig. 4e). Image analysis of picrosirius red staining revealed increased fibrillar collagen deposition in MCF-7:sh*CDH1* xenografts, validating that experimental *CDH1* down-modulation induced a functional EMT in vivo[16,29] (Fig. 4f, g). Molecular analyses confirmed reduced *CDH1* and increased *ZEB1* transcripts, while the expression of other EMT-TFs was unaffected (Fig. 4h).

*CDH1* knockdown reduced p120 and β-catenin proteins at cell junctions (Supplementary Fig. 6d). Ki67 and pHH3 indices were decreased whereas the apoptotic index was increased in MCF-7:sh*CDH1* compared to MCF-7:shSCR grafts (Supplementary Fig. 6e, f). The micro-metastatic load of mice engrafted with MCF-7:sh*CDH1* cells was around 10% of that of mice with MCF-7:sh*SCR* cells (Fig. 4i and Supplementary Fig. 6g). To exclude the possibility that the reduced metastatic burden merely reflected decreased primary tumor growth, we analyzed the micro-metastatic burden in mice with comparable primary tumor burden. In mice paired accordingly, MCF-7:sh*CDH1* cells caused smaller micro-metastatic loads (Supplementary Fig. 6h) indicating that decreased *CDH1* expression and the resulting EMP features do not increase the metastatic propensity of ER⁺ BC cells.

To exclude any potential systemic effects of genetically different tumor cells on the host, we grafted MCF-7:sh*SCR:GFP* and MCF-7:sh*CDH1:RFP* cells contralaterally (Supplementary Fig. 6i). The signal emanating from glands engrafted with MCF-7:sh*CDH1:RFP* cells at endpoint was lower than that from contralateral glands engrafted with MCF-7:sh*SCR:GFP* cells (Supplementary Fig. 6j). The number of sh*CDH1*:RFP⁺ lung foci was less than that of GFP⁺ loci and the total RFP⁺ area was smaller than the GFP⁺ area (Supplementary Fig. 6k,l). These results again support the hypothesis that E-cad down-modulation with associated EMP features does not favor metastasis in ER⁺ BC.

Similarly, MCF-7:*ZEB1* xenografts grew less and the engrafted glands weighed lower than MCF-7:*Ctrl* xenografts at four months (Fig. 4j, Supplementary Fig. 6m). Histology revealed that control cells invaded the muscle tissue adjacent to the thoracic mammary gland whereas ZEB1-overexpressing cells remained in situ with occasional microinvasion foci (Fig. 4k). The fibrillar collagen deposits were larger in MCF-7:*ZEB1* than in MCF-7:*Ctrl* xenografts (Fig. 4l, m). A trend for an increased apoptotic index was observed upon *ZEB1* overexpression (Supplementary Fig. 6n). *ZEB1* overexpression decreased levels of E-cad, p120, and β-Catenin proteins and occasional Vim⁺ cells were detected (Supplementary Fig. 6o). Again, the forced EMP state reduced the micro-metastatic burden (Fig. 4n, Supplementary Fig. 6p).

Overexpression of other EMT-TFs such as *ZEB2, TWIST1*, and *SNAI1* similarly reduced tumor growth and metastatic load (Supplementary Fig. 6q, r) arguing that different EMP states induced by distinct means in MCF-7 cells neither favor tumor progression nor metastasis.

## EMP in ER⁺ PDXs

To assess the relevance of these findings in MCF-7 cells to ER⁺ BC in general, we down-modulated E-cad expression in T99 and METS15 cells. In vivo growth (Fig. 5a) and mammary gland weight at end-point were reduced (Supplementary Fig. 6s). We confirmed reduced *CDH1* mRNA levels in the xenografted glands at endpoint (Fig. 5b). Furthermore, E-cad down-modulation reduced tumor invasion (Fig. 5c) and the micro-metastatic burden (Fig. 5d and Supplementary Fig. 6t). Similarly, ectopic *ZEB1* expression decreased T99 intraductal growth (Fig. 5e), the weight of xenografted glands (Supplementary Fig. 6u), and the micro-metastatic load in the lungs and bones (Fig. 5f, Supplementary Fig. 6v). Hence, an EMP per se, induced by genetic manipulation of individual genes, favors neither tumor progression nor metastasis of ER⁺ BC cells. This suggests that ER⁺ BC cells are plastic and the mesenchymal phenotype may be induced or stabilized after cells leave the primary tumor, possibly at the distant sites.

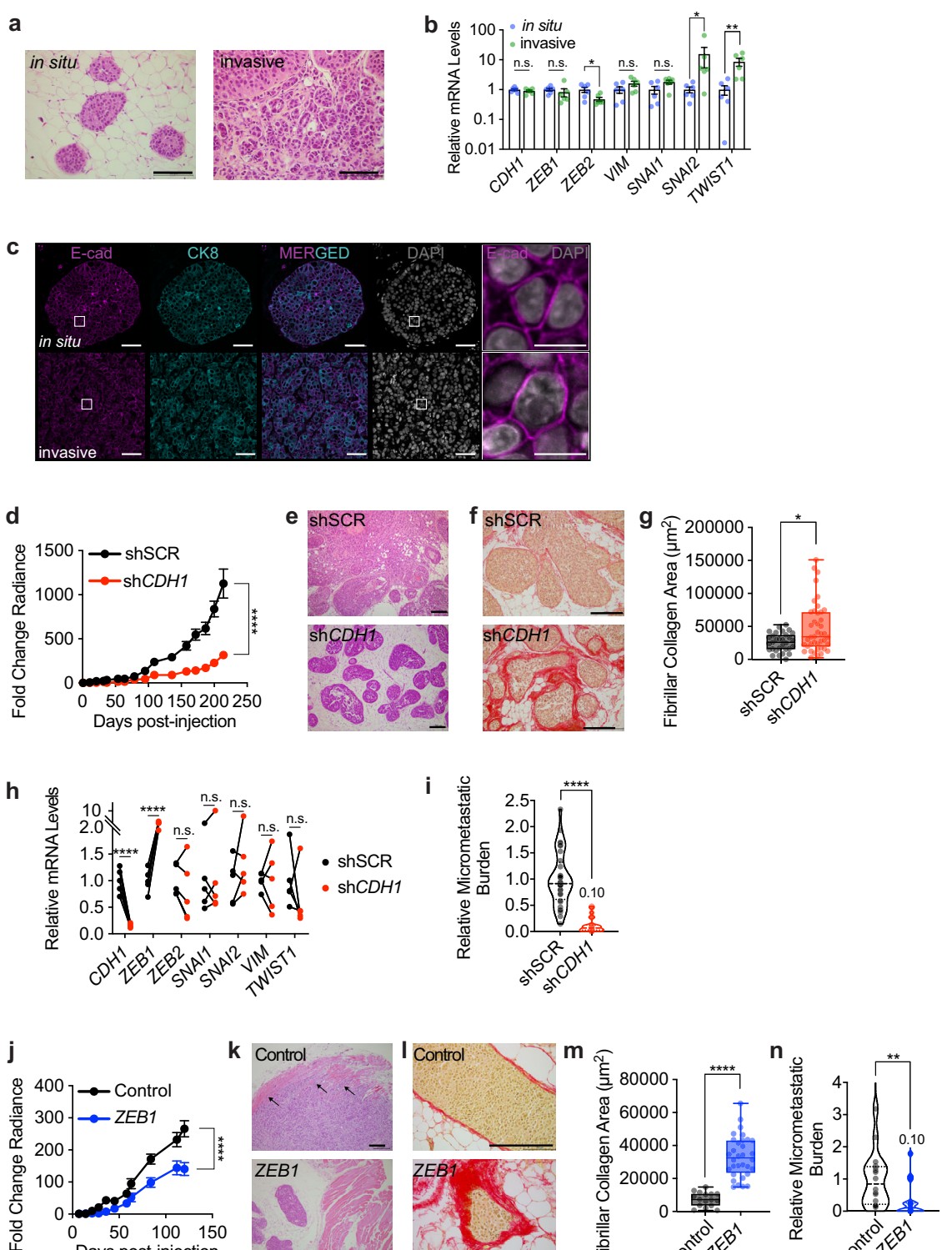

### E-cadherin is sufficient to awaken DTCs from dormancy

Our finding that dormant DTCs are in, at least partially, mesenchymal states, prompted us to assess whether returning to the epithelial state may reactivate proliferation. To this aim, we dissociated mammary glands and lungs from *NSG-EGFP* mice engrafted with MCF-7:*RFP* cells to single cells, plated them in 2D and applied drug selection to avoid overgrowth of mouse cells. RFP⁺ MCF-7 cells derived from primary tumors proliferated within a few days and were confluent by 1-2 weeks (Supplementary Fig. 7a). The lung-derived DTCs resumed proliferation after 2 months and formed epithelial islets (Supplementary Fig. 7b).

The brain DTCs, which had lower *CDH1* transcript levels than the lung DTCs (Fig. 3f), took longer to emerge from quiescence and formed epithelial islets at 4 months (Supplementary Fig. 7c). *CDH1* transcript levels were ultimately restored, and EMT-TFs, *ZEB1*, *ZEB2* and *VIM* transcripts decreased after serial passages in culture (Supplementary Fig. 7d–g), consistent with the hypothesis that reacquisition of an epithelial state enables cell proliferation.

To address whether restoring an epithelial state is sufficient to awaken dormant DTCs in vivo, we overexpressed E-cad in MCF-7 cells (Supplementary Fig. 7h). Bioluminescence one day after intraductal

**Fig. 4 | Role of EMP in ER⁺ tumor progression. a** Representative H&E micrographs of in situ and invasive MCF-7 intraductal xenografts, scale bars, 100 μm. **b** Relative mRNA levels of marker genes in in situ and invasive MCF-7 intraductal xenografts. Data represent mean ± SD from 6 glands in 3 mice, non-parametric Mann-Whitney test. **c** Representative E-cad IF micrographs in in situ and invasive MCF-7 intraductal xenografts, scale bars, 50 μm; inlet, 10 μm. **d** Fold-change radiance of MCF-7 shSCR and sh*CDH1* in intraductal xenografts, mean ± SEM, *n* = 19 or 20 xenografts from 5 mice/condition. Two-way ANOVA, multiple comparisons. **e, f** Representative micrographs of H&E (**e**) and sirius red (**f**) stained MCF-7.shSCR and sh*CDH1* xenografts from ≥3 mice. Scale bars, 200 μm. **g** Box plot showing the area of collagen deposition from *n* ≥ 31 ducts, ≥3 mice. Boxes span the 25th to 75th percentile, whiskers 1.5 times the interquartile range. Boxplot whiskers show minimum and maximum values. **h** Relative mRNA levels of *n* = 5 contralateral MCF-7:shSCR and sh*CDH1* xenografts, 5 mice. Paired t-test. **i** Relative micro-metastatic burden in 7 MCF-7 sh*SCR* and 8 MCF-7 sh*CDH1* tumor-bearing mice. Each dot represents a

single organ, the dashed line indicates the median and dotted lines indicate the lower and upper quartiles. Unpaired Student's t-test. **j** Fold-change radiance of control and MCF-7:*ZEB1* xenografts, data represent mean ± SEM, *n* = 16 glands, 4 mice each. Two-way ANOVA, multiple comparisons. **k, l** Representative micrographs of H&E (**k**) and sirius red (**l**) stained thoracic mammary glands bearing control or MCF-7:*ZEB1* cells, 3 mice each. Arrows indicate muscle invasion in thoracic mammary glands. Scale bars, 200 μm. **m** Box plot showing area of collagen deposition, *n* ≥ 17 ducts, ≥3 mice. Boxes span the 25th to 75th percentile, whiskers 1.5 times the interquartile range. Boxplot whiskers show minimum and maximum values. **n** Violin plot of the relative micro-metastatic burden from control and MCF-7:*ZEB1* intraductal xenografts-bearing mice. Dashed line indicates the median, dotted lines indicate the lower and upper quartiles. Unpaired Student's *t*-test. All mRNA expression levels were normalized to the geometric mean of *GAPDH* and *TBP*. *, **, ***, ****, and n.s. represent *P* < 0.05, 0.01, 0.001, 0.0001, and not significant, respectively.

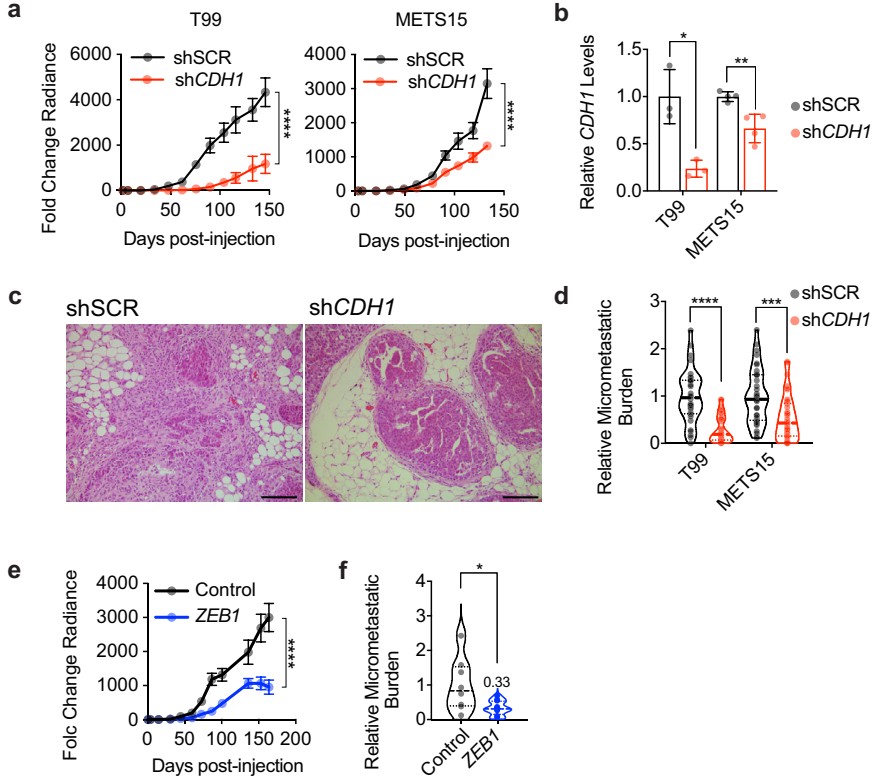

**Fig. 5 | Role of EMP in ER⁺ PDX progression. a** Growth curve of intraductal T99 and METS15 shSCR or sh*CDH1*. Data represent mean ± SEM of 16 xenograft glands for each shSCR and sh*CDH1* groups for T99 and 16 and 20 xenograft glands, respectively, for METS15, 5 mice for each cohort. Two-way ANOVA, multiple comparisons. **b** Relative *CDH1* expression in shSCR and sh*CDH1* intraductal xenografts (3 each for T99 and 4 each for METS15). Data represent mean ± SD. Unpaired Student's t-test. **c** Representative H&E micrographs of T99 shSCR and sh*CDH1* xenograft glands. Scale bar, 200 μm. **d** Violin plot of the relative micro-metastatic burden in T99 and METS15 intraductal xenografts-bearing mice. The dashed line indicates the median

and dotted lines indicate the lower and upper quartiles. Each dot represents a single organ. Unpaired Student's *t*-test. **e** Growth curve of vector control and *ZEB1*-overexpressing T99 intraductal xenografts. Data represent mean ± SEM of 15 and 18 xenograft glands, respectively. Two-way ANOVA, multiple comparisons. **f** Violin plot of the relative micro-metastatic burden in lungs and bones from 4 control and 5 *ZEB1*-overexpressing T99 xenografts-bearing mice. The dashed line indicates the median and dotted lines indicate the lower and upper quartiles. Unpaired Student's *t*-test. *, **, ***, ****, and n.s represent *P* < 0.05, 0.01, 0.001, 0.0001, and not significant, respectively.

engraftment was 2-fold higher in glands engrafted with MCF-7:*CDH1* cells compared to controls, suggesting that E-cad favors tumor cell survival and/or engraftment in the mouse milk ducts (Fig. 6a). Six months after injection, however, bioluminescence in MCF-7:*CDH1* grafts was 2-fold lower than in the controls (Fig. 6b). Similar to the 10-fold increase initially observed in vitro (Supplementary Fig. 7h), an 8-fold increase in *CDH1* transcript levels was observed (Fig. 6c). Ex vivo bioluminescence in the lungs from mice with MCF-7:*CDH1* xenografts was 10% of that of MCF-7:*Ctrl* xenograft bearing mice (Fig. 6d). Yet, a single lung lesion > 500 μm in diameter was detected

in 1 of the 4 mice engrafted with MCF-7:*CDH1* (Fig. 6e) when such a large lesion was never observed in over 60 mice engrafted with parental MCF-7 cells suggesting that E-cad expression may overcome dormancy.

To circumvent potential confounding effects of E-cad over-expression on engraftment and in vivo growth of MCF-7 cells at the primary site, we used a doxycycline (DOX) inducible E-cad expression vector (E-cad^IND) (Supplementary Fig. 7i). Five months after engraftment of MCF-7:*E-cad*^IND cells, when DTCs are readily detected in the lungs, a subgroup of mice was switched to DOX-containing

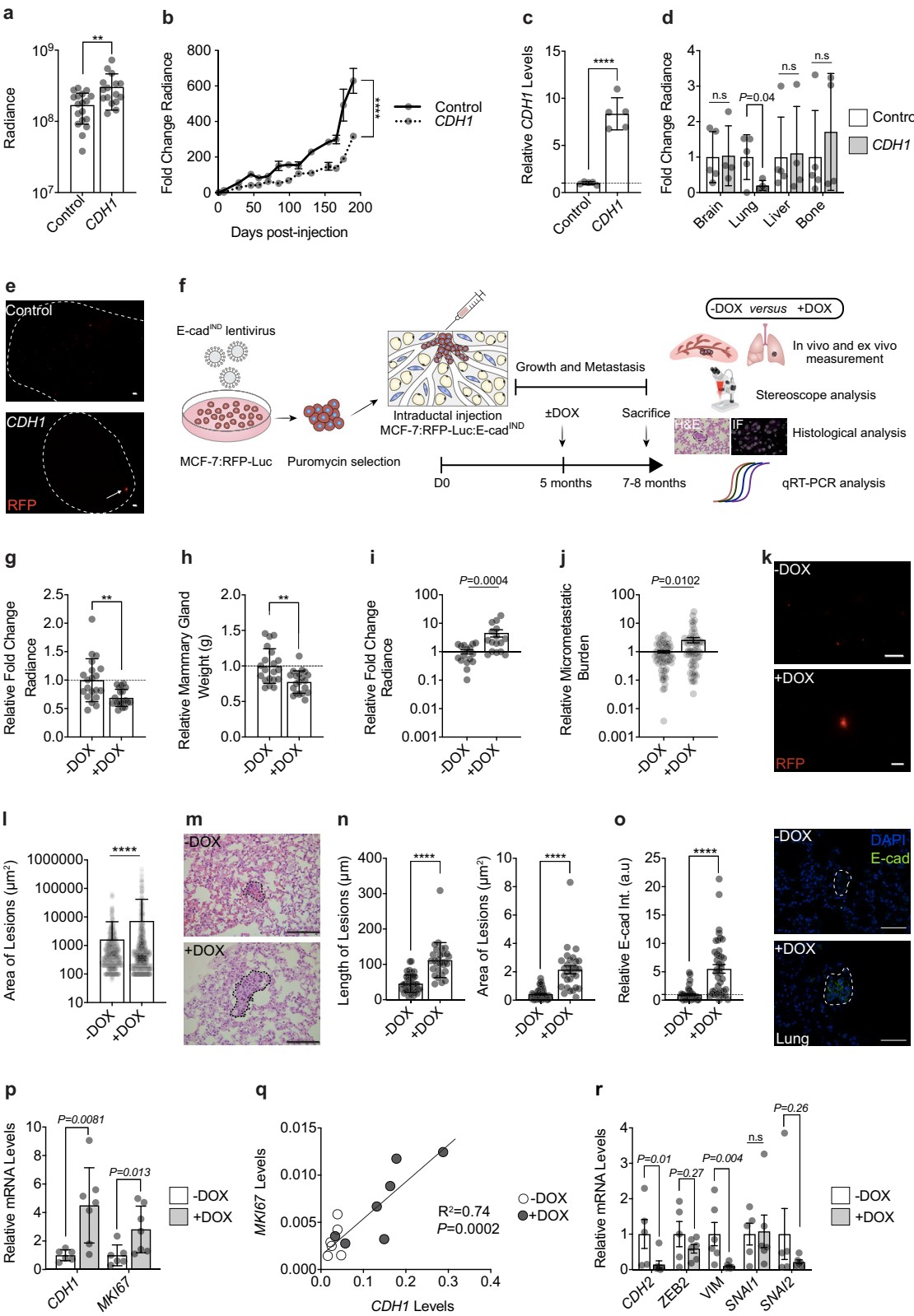

chow for 2-3 months (Fig. 6f). Bioluminescence and weight of mammary glands at sacrifice were significantly reduced in the DOX-induced group ($P < 0.01$) (Fig. 6g, h) but lung bioluminescence and the overall bioluminescence in distant organs were significantly increased (Fig. 6i, j). DOX treatment per se did not affect the weight of mice (Supplementary Fig. 7j), tumor growth, nor the weight of the engrafted glands nor the micro-metastatic burden at endpoint

(Supplementary Fig. 7k–m). Thus, ectopic E-cad expression is sufficient to drive DTCs out of dormancy.

To assess whether the increased lung bioluminescence was due to more seeding and/or increased size of individual lesions, we measured micro-metastases by fluorescence stereo microscopy and histology. *CDH1* induction increased the overall fluorescent area (Fig. 6k, l) as well as the area and length of lung lesions (Fig. 6m, n). The anti-E-cad IF

**Fig. 6 | E-cadherin in DTC awakening. a** Day 1 radiance of MCF-7 control and *CDH1* cells. Data represent mean ± SD, *n* = 16, 17 glands in 4, 5 mice. Unpaired Student's *t*-test. **b** Radiance fold-change. Data represent mean ± SEM, *n* = 16, 17 glands in 4, 5 mice **c** Relative *CDH1* mRNA levels in MCF-7 control and *CDH1* cells, *n* = 5. Data represent mean ± SD. Unpaired Student's *t*-test. **d** Ex vivo lung radiance. Data represent mean ± SD from 5 control and 4 *CDH1* mice. Multiple t-tests. **e** Fluorescence micrographs of MCF-7 control (top) or *CDH1* (bottom) lungs. Scale bar, 500 μm. **f** Experimental scheme. **g, h** Relative radiance and weight (**g**) of 20 xenografted glands from -DOX and +DOX mice. Data represent mean ± SD. Unpaired Student's *t*-test. **i** Fold-change lung radiance in - or +DOX, Data represent mean ± SEM, *n* = 16 in each group. Two-tailed Mann-Whitney test. **p ≤ 0.01. **j** Relative micro-metastatic burden. Data represent mean ± SEM, *n* = 69 from 16 mice/group. Nonparametric Mann-Whitney Test. **k** Representative fluorescence micrographs of lungs from - and +DOX mice. Scale bars, 500 μm. **l** Quantification of lung lesion area

from - or +DOX mice, *n* = 9 each. n≥476 lesions 9 mice/condition. Data represent mean ± SD. Unpaired Student's t-test. **m** Representative micrographs of lung sections from - or +DOX mice. Scale bar, 100 μm. **n** Quantification of lesion length and area in H&E stained lung sections from - and +DOX, *n* ≥ 30 lesions, 3 mice/condition. Data represent mean ± SD. Unpaired Student's t-test. **o.** Relative E-cad intensity (Int.) and representative IF of E-cad in lung lesions, *n* ≥ 38 from 3 mice/condition. Scale bar, 50 μm. Data represent mean ± SEM. Unpaired Student's t-test. **p** Relative *CDH1* and *MKI67* mRNA levels in lungs from 6 -DOX and 7 +DOX mice. Data represent mean ± SD. Nonparametric Mann Whitney test. **q** Pearson plot showing correlation between *MKI67* and *CDH1* mRNA levels in 6 -DOX and 7 +DOX mice. **r** Relative mRNA levels of EMT-TFs in lungs from 5 -DOX and 6 +DOX mice. Data represent mean ± SD. Nonparametric Mann Whitney test. All relative mRNA levels were normalized to *GAPDH*.

signal intensity of lung micro-metastases was 5.5-fold higher in +DOX than in -DOX mice (Fig. 6o). Semi-quantitative RT-PCR on lungs confirmed the induction of *CDH1* and revealed increased *MKI67* transcript levels in the +DOX group (Fig. 6p). *CDH1* expression correlated positively with *MKI67* expression (*P* < 0.001) (Fig. 6q), supporting that E-cad restoration drove cell cycle entry and proliferation. Finally, we asked whether the increase in proliferation in DOX-treated mice involved loss of mesenchymal phenotype. We noted a significant decrease in *CDH2* and *VIM* transcript levels (Fig. 6r), a trend to decrease *ZEB2* and *SNAI2* transcript levels, and unaltered *SNAI1* transcript levels (Fig. 6r). *ZEB1* was not reliably detected by the semi-quantitative RT-PCR approach (cycle threshold >37).

Similar experiments using T47D:*E-cad*^*IND* cells (Fig. 6f) showed that overexpressing E-cad did not significantly affect primary tumor growth nor mammary gland weight (Supplementary Fig. 7n, o) but increased the metastatic load 3-fold (Supplementary Fig. 7p). Analysis of fluorescence stereographs showed a 10-fold increase in area and fluorescence intensity of lesions upon E-cad restoration (Supplementary Fig. 7q–s). Thus, E-cad overexpression with resulting MET in two ER⁺ BC models is sufficient to drive DTCs out of dormancy.

### ESR1 mutations are not sufficient for awakening

*ESR1* mutations frequently occur in recurrences in patients treated with aromatase inhibitors[30]. To test the hypothesis that *ESR1* mutations may cause metastatic awakening, we xenografted MCF-7 cells with hotspot mutations *Y537S* or *D538G* knocked into the endogenous *ESR1* locus[31]. *ESR1*^*Y537S* did not affect primary tumor growth while *ESR1*^*D538G* promoted proliferation (Supplementary Fig. 8a). In neither case, however, metastasis to distant organs or the overall metastatic burden were affected (Supplementary Fig. 8b, c), indicating that these *ESR1* mutations were insufficient to awaken dormant MCF-7 cells in the MIND model.

### Genomic characterization of dormancy and associated pathways

To further investigate the exit from dormancy and to identify signals that drive it, we interrogated the transcriptome of DTCs. We dissociated mammary glands and lungs from mice bearing MCF-7:*RPF-Luc* intraductal xenografts for six months, FACS-sorted the single-cell suspensions based on RFP expression, and generated high-quality scRNA-seq profiles for >2,550 primary tumor cells and 194 DTCs.

Using the uniform manifold approximation and projection (UMAP) we identified eight clusters for primary tumor cells (Fig. 7a) and four for DTCs (Fig. 7b). Gene set enrichment analysis (GSEA) revealed a subpopulation of <0.6% primary tumor cells positively enriched for an EMT signature (Fig. 7c) while negatively enriched for E2F and MYC targets, MTORC1 signaling, ER responses, oxidative phosphorylation, and glycolysis (Supplementary Fig. 9a). The EMT cluster 7, showed decreased expression of epithelial/luminal genes *CDH1*, *EPCAM*, *ESR1*, and *KRT18* and increased expression of *VIM* and *FN1* as well as the dormancy gene *COL3A1*[32,33] (Fig. 7e). The two datasets

were stackable (Fig. 7d), suggesting that our experimental procedures did not introduce any major technical bias. DTC clusters 1 and 3 coincided with cluster 7 from the primary tumor. DTC clusters 1 and 3, henceforth called "inactive clusters", similarly lacked the expression of luminal epithelial markers and expressed *ZEB2* and *VIM* (*P* < 0.001) (Fig. 7f and Supplementary Fig. 9b). DTC clusters 0 and 2, henceforth called "active clusters", expressed *CDH1*, *EPCAM*, *KRT18*, and *ESR1* (Fig. 7f and Supplementary Fig. 9b).

The active clusters had positive GSEA scores for proliferation-related pathways, ER response, MTORC1 signaling, glycolysis, and oxidative phosphorylation (Supplementary Fig. 9c). IHC ER labeling index was reduced in lung DTCs compared to primary tumor cells (Supplementary Fig. 9d,e) suggesting decreased ER expression may result in reduced ER target gene expression. Cluster 2 contained some *MKI67*⁺ and *MCM2*⁺ cells while negatively enriched for an EMT signature (Fig. 7g). The inactive cluster 3 had increased expression of genes previously implicated in dormancy, including *ELN, COL1A1, COL3A1, FN1, BMPR2, and CXCL12* (Supplementary Fig. 9f). Together these findings suggest that DTCs have different degrees of EMP and dormancy, with cluster 3 representing the most dormant one.

These findings extended to the lung DTCs of METS15, in which *CXCL12, BMPR2, TGFBR2, FN1* transcripts were upregulated, and *COL3A1* to a lesser extent (Supplementary Fig. 9g). Experimentally induced EMP by *CDH1* knockdown or *ZEB1* overexpression upregulated *COL3A1* and *CXCL12* while ZEB2 overexpression increased *FN1* (Supplementary Fig. 9h).

To test the hypothesis that mesenchymal-like DTCs transit to an epithelial state during awakening, we assessed dynamic changes in mRNA expression with RNA Velocity, which models the instantaneous rate-of-change of RNA expression based on the balance of spliced and unspliced RNA molecules[34,35]. At the primary site the directionality of cells projected in the UMAP represented by velocity vectors varies between the subpopulations with the inactive cluster (7) not showing any directionality (Supplementary Fig. 10a). In lung DTCs, the velocity vectors of cells between the inactive and active clusters, at the "EMP bridge", point from the mesenchymal towards the epithelial clusters (Supplementary Fig. 10b). In addition, the phase plots of epithelial genes, *CDH1*, *EPCAM*, *ESR1*, and *KRT8* indicate transcript induction (Supplementary Fig. 10c) whereas those of the mesenchymal markers, *VIM*, *S100A4*, and *TCF4* show repression (Supplementary Fig. 10d) lending further support to the hypothesis that ER⁺ dormant DTC awakening requires a transition from a mesenchymal to an epithelial state.

To gain insights into the signaling pathways that control awakening, we compared global gene expression between the active and inactive clusters. As expected, GSEA for hallmarks showed that the active populations were enriched for DNA repair, oxidative phosphorylation, E2F targets, mTOR and estrogen responses whereas the inactive populations were enriched for EMT (Supplementary Table 4). IL6/JAK/STAT3 signaling and TNFα signaling via NFκB were positively

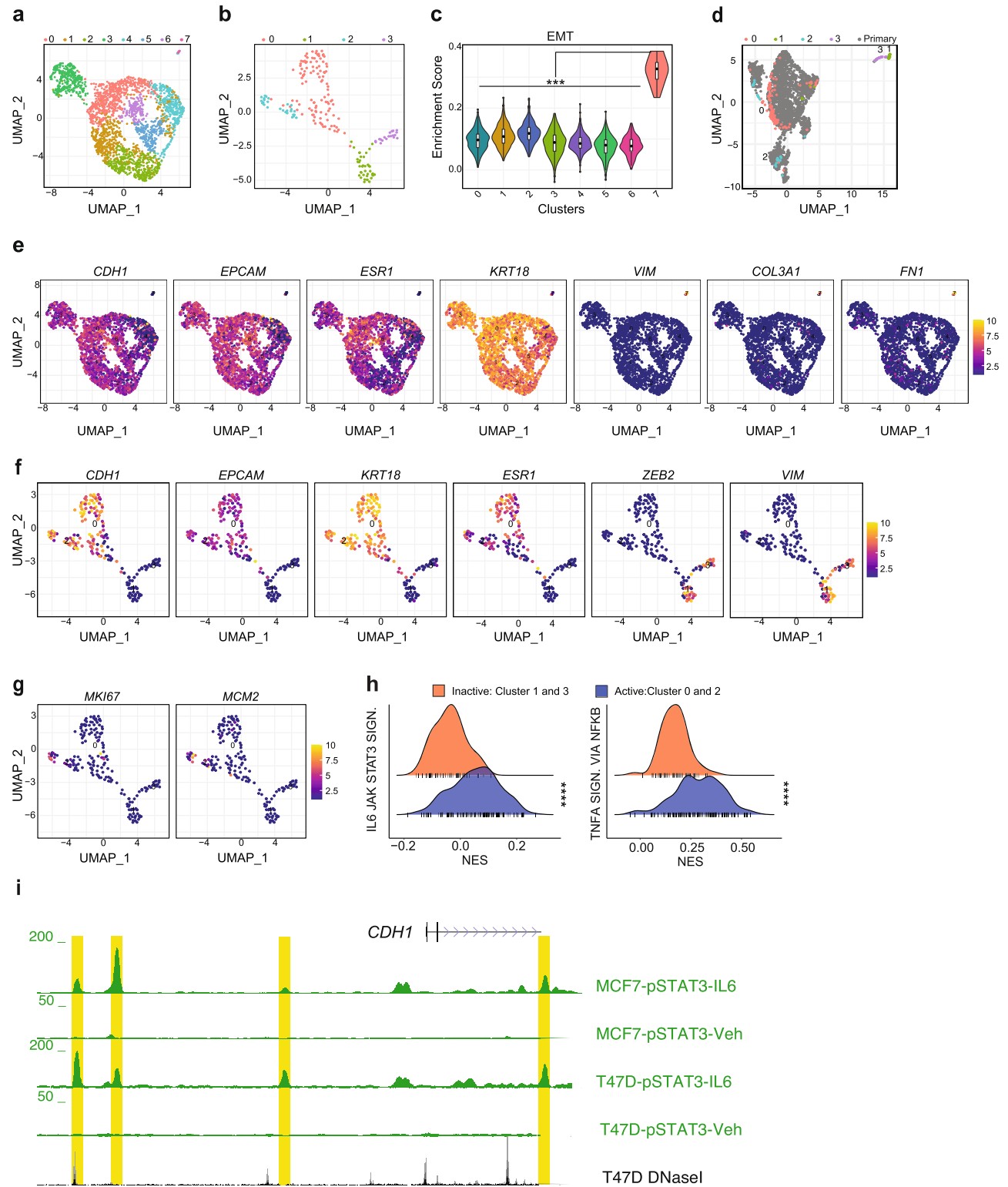

enriched in the active clusters (Fig. 7h), suggesting that cytokine sig-
naling, likely from the microenvironment, triggers tumor cell awa-
kening. Indeed, IL6-induced STAT3 activation increases metastatic
burden in T47D intraductally xenografted mice[36]. Analysis of pSTAT3
ChIP-Seq data from MCF-7 and T47D cells shows IL6-induced pSTAT3
binding to *CDH1* and *ESR1* regulatory regions (Fig. 7i and Supplemen-
tary Fig. 11a, b), in line with E-cad being a direct target of STAT3 acti-
vation that can determine awakening (Fig. 8).

## Discussion

Clinically, dormancy and late recurrence represent important chal-
lenges in ER⁺ BC[4,5,20]. Our understanding of the mechanisms controlling
these phenomena is limited largely due to the lack of reliable pre-
clinical models[7,8,9]. We show that MIND xenografts address this need;
they faithfully model the metastatic process of different BC subtypes;
whereas TN DTCs keep characteristics of their primary tumors, ER⁺
DTCs exhibit different states of dormancy.

**Fig. 7 | Heterogeneity of ER⁺ primary tumor cells and lung DTCs. a, b** UMAP plots representing FACS-sorted MCF-7:RFP⁺ cells isolated from primary tumors (**a**) and lungs (**b**). Colors code for different cell clusters. **c** Violin plot showing "HALL-MARK_EMT" gene set enrichment score in cell clusters of the primary tumors. Boxplots inside each violin describe the interquartile range, bold dots indicate median. Boxplot whiskers show minimum and maximum values. Statistical significance was assessed by fitting a generalized linear mixed model and computing the estimated marginal means across all the identified cell clusters. Tukey's method was used to assess adjusted p-values. **d** Integrated UMAP plot showing MCF-7:RFP cells isolated from primary tumors (grey) and lung tissues (color-coded according to clusters from b). **e–g** UMAP plots representing levels of selected transcripts in primary MCF-7:RFP (**e**) and lung DTCs (**f, g**). Numbers describe centroid of clusters identified in Fig. 7a and b. Violet gradient represents low, yellow gradient high gene expression. **h** Ridge plot showing normalize enrichment scores (NES) in the active versus dormant clusters identified in lung DTCs for the "IL6_JAK_STAT3_SIGNAL-ING" and "TNFA_SIGNALING_VIA_NFKB" hallmark gene sets. Statistical significance was assessed by Welch's t-test. **i** UCSC genome browser screenshot showing IL6-induced pSTAT3 binding at *CDH1* promoter and enhancer regions. The browser window shows regions, highlighted in yellow, which are bound by phosphorylated STAT3 upon IL6 stimulation in MCF-7 and T47D cells³⁶. *, **, ***, ****, and n.s represent *P* < 0.05, 0.01, 0.001, 0.0001, and not significant, respectively.

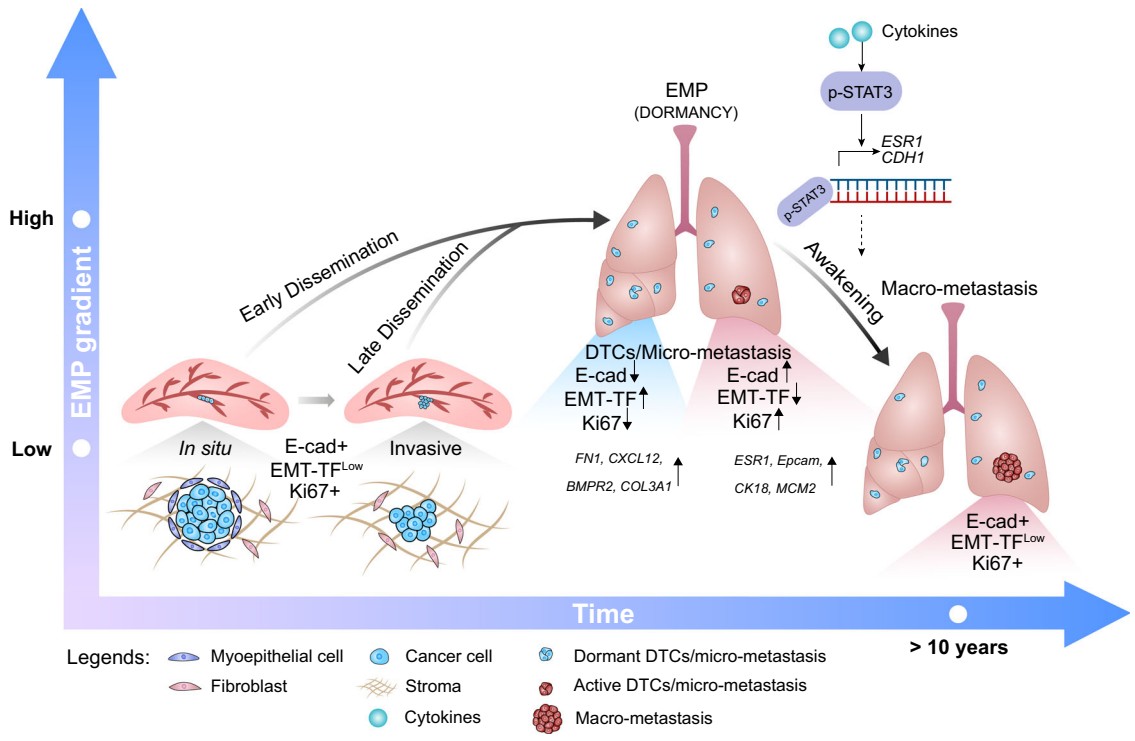

**Fig. 8 | Working Model for ER⁺ BC progression.** ER⁺ tumor cells progress from in situ to invasive stage in an epithelial state characterized by high expression of E-cad and low expression of EMT-TFs. At distant sites, DTCs show EMP features and enter dormancy. Awakening from dormancy involves restoration of E-cad expression that may be triggered by cytokines resulting in STAT3 activation and suppression of the EMT-TFs. Drugs promoting EMP or inhibiting the reacquisition of an epithelial state may prevent disease recurrence. EMT-TFs Epithelial-Mesenchymal Transition-activating transcription factors, EMP Epithelial-Mesenchymal Plasticity, DTCs Disseminated Tumor Cells.

We observed differences in the biology of ER⁺ versus TN DTCs which provide a possible explanation for different utilities of adjuvant chemotherapy between ER⁺ BC and TNBC. Adjuvant chemotherapy is in general successful in reducing TNBC recurrence because it can kill proliferating DTCs, and hence incipient metastases, whereas it provides limited benefit to ER⁺ BC patients because the target cells have low proliferative indices³⁷.

EMP, shown here to be characteristic of ER + BC dormant cells, has many important roles in tumor progression. It can drive invasion, migration, and metastasis³⁸. Our finding that reacquisition of an epithelial differentiation state is critical for the recurrence in ER⁺ BC (Fig. 8) is in line with a number of reports that the epithelial phenotype can instead be important for tumor cell dissemination, CTC survival, and metastatic outgrowth in different GEMMs³⁹⁻⁴³ and supports the notion that EMT may not always drive tumor progression but can have tumor type-dependent roles. Indeed, clinically, a partial EMT signature correlates with lymph node metastasis, reduced survival, and therapy resistance in head and neck carcinomas⁴⁴ but no such correlation is detected in other cancer types⁴⁵. Likely a partial EMT exists in many contexts but the specific genes and the strength of the signature as well as the frequency and its role in metastasis vary.

The upregulation of *SNAI2* and *TWIST1* transcripts in the invasive versus in situ disease is likely a consequence not a cause of invasion; we observed it also when ER + BC cells were directly injected into the fat pad¹⁴. In line with cell type-specific EMP states²⁵, our findings suggest that in ER⁺ human BC cells, the critical EMT-TFs are ZEB1 and ZEB2 whereas SNAI2 and TWIST1, critical for EMP in other tissues, control a distinct cellular differentiation program linked to basal features¹⁴. While we find *ZEB1/ZEB2* transcripts increased in all organs sites, further experiments are needed to determine organ-specific aspects of the EMP.

Our finding that both E-cad upregulation and down-modulation reduced the growth of primary tumors suggests that the E-cad levels need to be tightly regulated. The E-cad down-modulation observed in dormant DTCs is likely due to EMT-TFs directly binding the *CDH1* promoter and repressing its activity⁴⁶,⁴⁷. The observation that ER⁺ PDXs, whether established from a primary tumor (T99) or a metastatic lesion (METS15), disseminate and enter dormancy suggests that these

processes are not genetically but epigenetically controlled. Distinct epigenetic states may also account for the a priori surprising finding that cytokine signaling pathways previously implicated in dormancy induction may activate awakening. The transcriptional outcomes, and hence pro- versus anti-EMP effects, are likely chromatin context dependent. We have previously demonstrated that ER signaling is growth promoting during puberty but inhibitory during pregnancy in normal mouse mammary epithelial cells[48] suggesting that epigenetic plasticity is an intrinsic property of ER⁺ breast epithelial cells, be they normal or transformed.

A limitation of the MIND models used in the present study is the lack of an intact immune system in the host mice. The awakening process has been shown to involve multiple cell types[33,49–51] and secreted factors. Although NSG mice have some innate immune cells, functions of these cells might not be normal in the absence of B and T cells. Whether active immune cell signalings regulate the expression of *CDH1* or others also needs to be determined. This can ultimately be addressed by restoring a functional human immune system in host mice[52].

We failed to detect different EMP states in primary tumor cells. However, a minor cell cluster has a positive EMT enrichment score, with increased *VIM, COL1A1, COL3A1, COL12A1*, and *FN1*, and decreased *CDH1* and proliferation markers. We were unable to unequivocally conclude whether these rare cells may be at the origin of DTCs because of our inability to isolate them based on specific cell surface markers for FACS-based purification. Given that the experimental induction of EMP did not increase metastatic dissemination and progression, we favor the hypothesis that the mesenchymal state is induced at the distant site. Barcoding experiments will help to distinguish between the two scenarios.

We present evidence that mesenchymal lung DTCs transit into an epithelial cell state using RNA velocity. This approach is intended for predicting cell states within a number of hours, likely a much shorter time-scale than the in vivo plasticity observed in patients. Yet, we are studying these DTCs at extremely late stages, where a number of cells are already transitioning from M to E. In addition to this, we cannot exclude that the majority of mesenchymal DTCs may have been lost during tissue dissociation. In order to distinguish between the two scenarios further studies would be required for example repeating these experiments using a lineage tracing barcoded library in combination with scRNA-seq.

The present findings suggest that stabilizing a mesenchymal cell state is critical to prevent recurrence. EMT-TFs, ZEB1/ZEB2, SNAIL, and TWIST, as well as EMT-inducing factors, such as TGFβ, have all been shown to halt the cell cycle by decreasing Cyclin D1 protein levels and increasing the phosphorylation of Rb[53–56]. The tight link between EMT and cell cycle control suggests the intriguing possibility that underlying the success of CDK4 inhibitors may be a block of exit from dormancy. This and other hypotheses are now readily amenable to experimental testing with intraductal xenograft models.

## Methods

### Clinical samples

The Commission cantonale d'éthique de la recherche sur l'être humain (CER-VD 38/15, PB_2016-01185) approved this study, informed consent was obtained from all subjects. Tumor tissue was mechanically and enzymatically digested using parallel razor blades and collagenase, as previously described[13,14]. Effusion samples (pleura or ascites) were centrifuged at 380 × g at 25 °C for 10 min. Pellets were rinsed with phosphate-buffered saline (PBS), 2% calf serum (CS), and erythrocyte-lysed using red blood cell lysis buffer (Sigma, R7757) for 5 min, then diluted in PBS 2% CS, and centrifuged again. Patient-derived tumor cells were transduced with either ffLuc2/Turbo-GFP or ffLuc2/Turbo-RFP (GEG-tech) overnight in low attachment culture plates (Corning® Costar® Ultra-Low Attachment) in a humidified incubator (37 °C, 5% $CO_2$, and 5% $O_2$).

### Animal Experiments

All mice were maintained and handled according to Swiss guidelines for animal safety with a 12-h-light-12-h-dark cycle, controlled temperature (22 ± 2 °C) and humidity (55 ± 10%), and food and water *ad libitum*. Experiments were performed in accordance with protocol VD1865.5 approved by the Service de la Consommation et des Affaires Vétérinaires, Canton de Vaud, Switzerland. NOD.Cg-*Prkdc^scid Il2rg^tm1Wjl*/ SzJ (NSG) and NOD.Cg-*Prkdc^scid Il2rg^tm1Wjl* Tg(CAG-EGFP)1Osb/SzJ (NSG-EGFP) mice were purchased from Charles River and The Jackson Laboratory. For MCF-7 intraductal xenografts, the maximal tumor radiance is $2 \times 10^{11}$ p/sec/cm²/sr, while for the rest of the intraductal xenografts, the maximal tumor radiance is $1 \times 10^{10}$ p/sec/cm²/sr or 1 cm³. The maximal tumour size/burden was not exceeded. Most experiments were conducted using the NSG strain, unless stated otherwise in text. For the induction of E-cad, doxycycline (0.62 g/kg of food, SAFE 150 SP-25 www.safe-diets.com) was administered in the diet.

### Intraductal injections and re-transplantation

Mice were anesthetized by intraperitoneal injection of 200 μl of 10 mg/kg xylazine and 90 mg/kg ketamine (Graeub). Intraductal injections of single-cell suspensions were performed as previously detailed[14,57]. In all mice experiments, intraductal xenografts of cell lines and patient-derived cells were generated by injecting $1 \times 10^5$ and $2 \times 10^5$ cells, respectively, into the teats of 8-16-week-old *NSG* or *NSG-EGF* female mice, and grown for 4–6 months. For re-transplantation of patient-derived cells, mammary glands were collected on ice-cold 1X PBS, dissociated using parallel razor blades, and enzymatically digested using the tumor dissociation kit (Miltenyi Biotec) to generate single cells (human and mouse). To enrich for human cells the mouse cell depletion kit (Miltenyi Biotec) was used according to manufacturer's instructions. Cells were counted and $2 \times 10^5$ cells were intraductally injected.

### Cell culture, cloning, and cell growth

ER⁺ BC cell lines MCF-7 and T47D as well as TN BC cell lines BT20 and HCC1806 were purchased from American Type Culture Collection (ATCC). MCF-7, T47D, BT20, and HCC1806 were maintained in Dulbecco's modified Eagle's medium (DMEM) medium (cat# 31966, Gibco) supplemented with 10% FCS (cat# 10270-106, Thermo Fisher Scientific Inc.) and penicillin/streptomycin (P/S, cat# 15070-063, Thermo Fisher Scientific Inc.). Parental and *ESR1*^mutant (*Y537S* and *D538G*) MCF-7 cells were transduced with either ffLuc2/Turbo-GFP or ffLuc2/Turbo-RFP, and selected for the brightest fluorescent subpopulation by FACS sorting. shFF3 and sh*ZEB1*, PstByGFP and PstBy-Zeb1 from Sendurai Mani (MD Anderson). Inducible E-cad vector was generated by annealing the ORF-*CDH1* (Clone ID:IOH46767, Invitrogen) cassette into an inducible backbone (pLIX403, cat# 41395, Addgene, https://www.addgene.org/41395/) using the Gateway cloning strategy (cat# 12535, Invitrogen). Control eGFP, *ZEB2, TWIST-1*, and *SNAI1* overexpression vectors from VectorBuilder (https://en.vectorbuilder.com/). pLKO.1 puro shRNA E-cad was a gift from R.A. Weinberg (Addgene plasmid # 18801) pLKO.1 puro, pLKO.1 sh*CDH1*[58] were purchased from Addgene. pLL3.7m-Clover-Geminin(1-110)-IRES-mKO2-Cdt(30-120) was a gift from Michael Lin (Addgene_83841; http://n2t.net/addgene:83841; RRID:Addgene_83841). Lentiviruses were produced as described[59], lenti-ONE GFP or RFP:Luc were designed by and purchased from GEG Tech.

### Tumor growth and metastasis analysis

Tumor growth was monitored every other week by in vivo imaging system (IVIS, Caliper Life Sciences). Briefly, 12 min after intraperitoneal administration of 150 mg/kg luciferin (cat# L-8220, Biosynth AG), mice were anesthetized in an induction chamber (O₂ and 2% isoflurane) and placed inside the machine. Images were acquired and analyzed using

Living Image Software version 4.4 (Caliper Life Sciences, Inc.). For ex vivo bioluminescence measurements, mice were first injected with 300 mg/kg luciferin for 7 min, then injected with 1 ml of 10 mg/kg xylazine and 90 mg/kg ketamine. Resected organs were then imaged 20 min after luciferin injection. Mammary glands were fixed in 4% paraformaldehyde (cat# 0335.3, Carl Roth) overnight at 4 °C or flash-frozen in liquid nitrogen for RNA or protein extraction.

## Immunohistochemistry and passive clarity

Histological staining was performed as detailed previously[60]. For passive clarity, tumor-bearing mammary glands and lungs, were embedded in 5% agarose (w/v, PBS, cat# 16500500, Invitrogen), left at room temperature to solidify and agarose cubes then were removed from the plastic container and mounted for Vibratome sectioning (Leica VT1200 S) using glue (cat# 14460, Ted Pella, Inc.). The buffer tray was then filled with cold PBS. Blade travel speed was 0.8 mm/sec and minimum thickness of sections was 0.5 mm. To preserve endogenous tissue fluorescence, Rapiclear 1.52 solution (RC, SunJin Lab, https://www.sunjinlab.com/) was used according to the manufacturer's instructions with some modifications. Tissues were permeabilized in 2% v/v Triton X-100 (Sigma T8787) PBS overnight on a gentle rotor shaker at room temperature. Rapiclear 1.52 was pre-warmed at 37 °C and 2 ml were added on top of the tissues for 1 hr. Tissues were mounted between 2 coverslips using RC 1.52 and iSpacers for image acquisition and long-term storage. Z-stacks were acquired using a Zeiss LSM700 confocal microscope and 3D-reconstructed using Imaris bit-plane and Fiji image analysis software (version 1.52t). The list of primary and secondary antibodies is provided in Supplementary Table 2.

## Immunohistochemistry analysis

For analysis of primary tumor staining, outlines of human cells were drawn manually with "Freehand selections" and a built-in function "waitForUser" for each image to exclude mouse cells, followed by ROIs creation. Channels were then binarized using "Huang" thresholding algorithm and images were processed with watershed segmentation to split closely touching cells and then denoised by a Minimum Filter with radius of 1 pixel. Cell numbers were quantified using the "analyse particles" function of Fiji (Size 0.01-infinity pixel$^2$, circularity 0-1), channels were saved, cell quantifications were output as a table for further data analysis. For analysis of metastatic BC cells in lungs, human cell nucleus and Ki67$^+$ cells were counted manually.

For analysis of the FUCCI reporter in primary tumors, images were analysed every 8 slices from the original Z-stack images to meet the criteria of the Nyquist–Shannon sampling theorem. Green and red channels of interest were binarized using the "Li" thresholding algorithm chosen based on visual inspection of output images. Images were processed with watershed segmentation to split closely touching cells and denoised by a Minimum Filter with radius of 1 pixel. The masks of the two channels were used to generate a third mask using the "AND" operation from "Image Calculator" to indicate double positive cells. Cell numbers were quantified using the "analyse particles" function of Fiji for all 3 masks (size 1-infinity pixel$^2$, circularity 0-1) and the outlines of quantified cells were recorded in another three masks (Show = outlines). Finally, the six masks (two channels of interest, one from the AND operation and their 3 masks showing outlines) were merged and saved for visualization and quality inspection. For analysis of FUCCI reporter in lungs, green, red and double positive cells were quantified manually. The percentages of cycling and non-cycling cells were quantified using the following equations:

$$\text{Cycling cells}(\%) = \frac{\text{No. of green cells}}{\text{No. of (green cells + red cells − double positive cells)}} \times 100\% \qquad (1)$$

$$\text{Non − cycling cells}(\%) = 100\% − \text{Cycling cells}(\%) \qquad (2)$$

For analysis of mesenchymal-like cells in primary tumors, the blue channel of interest (DAPI) was binarized using the "Percentile" thresholding algorithm for MCF-7 and T47D and "Moments" for METS15 tumors based on the visual inspection of output images. Images were processed with watershed segmentation to split adjacent cells, followed by cellular aspect ratio (CAR) quantification using the "analyse particles" function of Fiji (Size 50-infinity pixel$^2$, circularity 0-1). Mask for the blue channel was saved and the cell quantifications were output as a table for further data analysis. For analysis of mesenchymal-like cells in lungs, major and minor axis were drawn manually on each tumor cell for CAR quantification, >300 DTCs in lungs from each mouse, $n = 3$. Mesenchymal-like cells were defined as those with major axis:minor axis ratio (CAR) > 1.6. For analysis of fluorescence intensity, corrected total cell fluorescence (CTCF) was calculated by manually drawing an outline around the border of the cells using the drawing/selection tools in ImageJ, and a close-by region with no cells to serve as a measure of background noise. Measurements were generated by ImageJ and CTCF was calculated as: CTFC= integrated density-(area x mean fluorescence background). For analysis of fibrillar collagen deposition, area of collagen was automatically quantified using ImageJ using size filter to exclude cells within the ducts.

Fluorescence images of whole mammary glands and lungs were acquired with a LEICA M205FA fluorescence stereo microscope equipped with a Leica DFC 340FX camera.

## RNA extraction and RT-PCR

Cell pellets, tumor-bearing mammary glands, and microdissected lung, liver, and brain micrometastases were homogenized with TRIzol reagent (Invitrogen), total RNA was isolated with miRNeasy Mini Kit (Qiagen), cDNA was synthesized with random p(dN)$_6$ primers (Roche) and MMLV reverse transcriptase (Invitrogen). RT-PCR analysis in triplicates was performed with SYBR Green FastMix (Quanta) reaction mix and analyzed on QuantStudio 6 and 7 Flex Real-Time PCR System Software. Supplementary Table 3 provides the list of primers.

## Immunoblotting

Cell pellets and tumor-bearing mammary glands were homogenized in 1% SDS solution containing protease (cat# 11836153001, Roche AG) and phosphatase inhibitor cocktail (cat# 04906845001, PhosStop, Roche, AG), followed by 20 min centrifugation at 4 °C (14,000 x g). Protein-containing supernatant was quantified and normalized to reference protein using the Image Studio Lite version 5.2 (LI-COR®). SDS-polyacrylamide and transfer as previously detailed[14], membranes were revealed with ECL or WesternSure PREMIUM Chemiluminescent Substrate (cat# 926-95010, LI-COR®). Antibodies are listed in Supplementary Table 2.

## Single cell RNAseq and RNA velocity

Intraductal xenografts of MCF-7:*RFP-Luc* were grown for 6 months. Mammary glands and lungs from 2 *NSG-EGFP* mice were collected and were mechanically and enzymatically digested using parallel razor blades and 1.5% Collagenase A/1% hyaluronidase, respectively. Samples were centrifuged at 380 x g at 25 °C for 10 min, and resuspended in red blood cell lysis buffer (Sigma, R7757) for 3–5 min, then diluted in PBS 2% CS, and centrifuged. Pellet was resuspended in trypsin for 3 min at 37 °C, centrifuged and resuspended in 0.1 mg/ml DNase (1284932, Roche AG). After centrifugation, pellet was resuspended in 10 ml PBS 2% CS, filtered through a 40 μm pore size filter (cat#352350, BD Falcon) for FACS. GFP$^-$RFP$^+$DRAQ7$^-$Hoechst 33342$^+$ cells were sorted

using BD FACSAria-II SORP and analyzed on the BD FACSDiva software version 9.0 (Supplementary Fig. 12a, b).

Sorted cells were washed and resuspended to 1000 cells/μL in PBS. Single-cell RNA sequencing was performed using 10x Genomics Chromium V3.1 kit, and Illumina Hiseq 4000. After demultiplexing sequencing libraries to individual FASTQ files, Illumina output sequencing reads from 10x Genomics Chromium were processed and aligned to the GRCh38 genome using "count" function of Cell Ranger v5.0.1. Quality controls were performed >500 genes detected per cell and <20% mitochondrial reads. Seurat software v4.0 default settings were used to ensure conservative clusters. The number of principal components chosen to capture the majority of the variation in the data was based on elbow plots. GSEA were performed using the "enrichIt" function of "escape" version 1.6.0 in R. The hallmark gene collection MSigDB v6.2 was used.

The *scvelo* algorithm version 0.2.4 was used to compute the steady-state model of RNA velocity using the tool sc.tl.velocity[35]. To fit the slope of the steady-state model 92nd instead of 95th percentile was used to better capture the steady-state. RNA velocity vectors were projected onto UMAP plot using default settings, single genes were interrogated using phase plots available in the package. For primary tumour cells, the scvelo package was used to calculate velocities on the deterministic setting with default parameters and to generate UMAP plots and the corresponding low-dimensional velocity embeddings.

### ChIP-seq track visualization
The genome browser screenshots shown in Fig. 7i and Supplementary Fig. 11a, b were generated by accessing the USCS genome browser via the BC epigenomics track (BC) Hub, (https://bchub.epfl.ch) (bioRxiv 2022.05.01.490187).

### Statistics and illustrations
Statistical analysis was performed using GraphPad Prism (version 8) (San Diego, California, USA, www.graphpad.com). Statistical tests are indicated in the figure legends. For growth curves, two-way ANOVA with multiple comparisons was used, for contralateral intraductal grafts, paired Student's t-test was performed, and for individual grafts, unpaired Student's t-test were performed, while testing for normality (Shapiro-Wilk test). Non-parametric statistical tests were used when Kolmogorov-Smirnov and Shapiro-Wilk normality tests failed to show a normal distribution. All statistical tests are two-tailed. *, **, ***, ***, and n.s. represent $P < 0.05$, 0.01, 0.001, 0.0001, and not significant, respectively. Graphic rendition of stereoscope in Fig. 6f was created with BioRender.com.

### Reporting summary
Further information on research design is available in the Nature Research Reporting Summary linked to this article.

## Data availability
The raw and processed scRNA-seq generated in Fig. 7 and Supplementary Figure 9 have been deposited in the Gene Expression Omnibus (GEO) database under accession code GSE196936. All data for the main and supplementary figures are available within the article, in the source data file section. Source data are provided with this paper.

## Code availability
The code employed for the analyses is open-source and available through the above mentioned packages in R.

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

## Acknowledgements

We thank S. Egan for critical comments and P. den Hollanader for careful reading of the manuscript, J. Aguirre-Ghiso, R. Siersbæk, I. Tirosh, G. LaManno and N. Aizarani for helpful discussions, L. Battista and Y. Liu for technical assistance, J. Dessimoz, EPFL histology core facility, T. Laroche, EPFL bioimaging and optics platform (BIOP), M. Garcia at the EPFL and D. Labes at AGORA flow cytometry facilities and B. Mangeat at the EPFL gene expression core facility (GECF) for technical assistance. We thank students G. Pittet, J. Fan, M. Lasfargues, M. Levorato, J. Brune, A. Huguenin-Dumittan, and A-M. Curat and the patients who participated in our study. P.A., H.Q. and F.D.M were supported by SNF 310030_179163/1 Exploring key steps of the metastatic cascade in ER⁺ BC in vivo, Y.Z. by KFS-4738-02-2019-R Different facets of estrogen receptor alpha (ER) signalling during ER⁺ breast carcinogenesis, and G.S. by Biltema ISREC Foundation Cancera Stiftelsen, Mats Paulssons Stiftelse, and Stiftelsen Stefan Paulssons Cancerfond. H. Q. is supported by Deutsche Krebshilfe. C.B. has support from and K.M. is supported by H2020-MSCA-ITN (ITN-2019-859860-CANCERPREV). This manuscript was edited at Life Science Editors.

## Author contributions

Conceptualization P.A., C.B.; Formal Analysis P.A., Y.Z., F.D.M, G.A., K.M., C.S.; Investigation P.A., Y.Z., C.S., H.Q., G.S.; Resources S.M., S.A.; Writing P.A., C.B; Funding Acquisition C.B.

## Competing interests

The authors declare no competing interests.
