## [Peer Review File · Nature Communications]

Epithelial-mesenchymal plasticity determines estrogen receptor positive breast cancer dormancy and epithelial reconversion drives recurrenceREVIEWER COMMENTS

Reviewer #1 (Remarks to the Author):

The manuscript entitled "Epithelial-mesenchymal plasticity determines estrogen receptor positive (ER+) breast cancer dormancy and reacquisition of an epithelial state drives awakening" provides comprehensive data from human ER+ and TN breast cancer cell lines and PDX cells injected into the mammary ducts of immune deficient mice to support the relevance of epithelial-mesenchymal plasticity in tumor dormancy and metastatic recurrence.

In this study, authors observed that ER+ disseminated tumor cells (DTCs) exhibit lower proliferative indices when compared to TN DTCs, which correlates with a delayed metastatic recurrence. In this model, metastatic dormancy is reversed by the induction of CDH1 expression in ER+ DTCs. Using breast tumor cell lines and PDX models in the study complement each other to emphasize its clinical relevance to the late recurrence that characterizes ER+ breast cancer patients. The in vivo characterization of tumor models used are carefully analysis with detailed quantification. However, several key issues should be addressed to provide further support to the conclusion the authors intend to draw.

1. A major conclusion that the authors intend to draw is that induction of EMP does not promote tumor cell dissemination in ER+ tumors. The key data presented to support this conclusion is Fig. 4 showing that knocking down CDH1 or expression of ZEB1 inhibited primary tumor invasion and tumor dissemination in the lung and the bone. In contradiction with this study, CDH1 downregulation or Zeb1 overexpression are usually associated with increased invasive capability in tumor cells. While the authors show a very limited marker changes associated EMP in Fig. S4, it is not convincing whether knocking down E-cad or expression of ZEB1 indeed activates a functional EMT program. Instead, knocking down CDH1 and overexpression of ZEB1 could lead to apoptosis and reduced cell proliferation in various cell types, which could explain the reduced primary tumor growth and the benign tumor morphology presented in tumors with CDH1 knockdown or Zeb1 overexpression. Further functional characterization of the effect of CDH1 knockdown or Zeb1 overexpression on cell proliferation, apoptosis, adherents junction integrity, cell migration and invasion should be carried out in culture. Furthermore, in Fig. 4C, E-cad staining signal is very weak and hard to see whether E-cad is on the adherents junctions. For both primary tumor with CDH1 knockdown or Zeb1 overexpression, apoptosis markers and additional EMP markers, including E-cadherin, beta-catenin, vimentin and desmosome markers should also be stained to evaluate the EMP status of these tumor cells.

2. Since this conclusion that EMT inhibits tumor invasion and dissemination is contradictory to vast amount of published data demonstrating that induction of EMT promotes tumor cell invasion and dissemination into the circulation, additional approaches, such as TGF-beta treatment and induction of Snail1 or Twsit1 should be used to trigger EMT and test their impact on MCF7 tumor invasion and dissemination. The reason to do so is because that knocking CDH1 in epithelial tumor cells is not equivalent to activation of EMT since loss of CDH1 without concurrent activation of other mesenchymal program could lead to severe apoptosis. Similarly, Zeb1 activation alone may not be sufficient to induce EMT and might lead to apoptosis and growth inhibition. Therefore, additional approaches are needed to support this conclusion.

3. Fig. 5 clearly demonstrated that DTCs of MCF7 tumors in distant sites present a more mesenchymal phenotype with endogenously induced EMT-TFs. But it is unclear when EMT is activated during MCF7 cell dissemination from primary tumors to distant sites. The conclusion that EMT didn't occur in primary tumors is indirectly deduced based on the data from CDH1 knockdown and Zeb1 overexpression. To support such conclusion, it is important to examine primary tumors to detect such rare tumor cells that have undergone EMT. Especially, Yu et al. (2013, Science reported that even ER+ breast cancer patients have CTCs that present a mesenchymal molecular signature. Therefore, it is also important to examine circulating tumor cells from the blood in MCF7 tumor-bearing mice for EMP features.

4. The authors determined that dormant ER+ DTCs presented mesenchymal features (Fig. 3) and inducing a transition to more epithelial phenotypes through E-cadherin overexpression promotes their growth in the secondary site (Fig. 5). Authors made opposite observations in TN DTCs, which

rapidly progressed to macrometastatic lesions (Fig. 1). To further support their model, authors should determine whether rapid macrometastatic outgrowth detected in TN breast cancer cell lines is associated to the presence of epithelial phenotypes and mesenchymal-epithelial transition in the secondary site.

5. Suppl Fig. 5a-d shows that lung and brain-derived DTCs restore their proliferative ability when cultured in vitro. This enhanced proliferation in vitro correlates with changes in CDH1 and ZEB1 expression, in accordance with the hypothesis that reacquisition of epithelial phenotypes promotes cell proliferation. However, some of the EMT-TFs analyzed in fig. 5b show some inconsistency with this observation. Concretely, SNAI2 expression is increased by 10-fold in Brain DTCs cultured in vitro, and ZEB2 is increased by 5- and 50-fold in lung and brain DTCs, respectively. Since these changes are much bigger than the ones observed in CDH1 and ZEB1 expression, authors should evaluate the relevance of downregulating expression of each EMT-TF in to see which are more relevant for cell proliferation.

Technical points:

1. The authors state that MCF-7:shSCR cells invaded the stroma whereas the MCF-7:shCDH1 cells remained in situ, providing images of the two conditions in figure 4F. However, the image of MCF-7shSCR condition does not show the edge of the tumor nor stromal infiltration but only a tumor that is much bigger in size.

2. Data represented in Figure 4D regarding the expression of androgen, estrogen and progesterone receptors requires more explanation in the main text.

Reviewer #2 (Remarks to the Author):

Aouad and colleagues investigate mechanisms determining tumor cells dormancy and reawakening in mouse models using intraductal injection. They find a link between epithelial-mesenchymal plasticity (EMP) and induction of dormancy, and observe that E-Cadherin forced expression enables to overcome dormancy and promote metastasis growth. While the authors should be commended for their focus on this highly relevant topic (DTCs), altogether, this study suffers from several major weaknesses:

(1) The authors use different strategies to determine whether or not (and how many) DTCs are dormant (proliferative index, FUCCI, p27 staining). While Ki67/proliferative index mainly tells about proliferative rates and not necessarily about dormancy, FUCCI highlights cells at different cell cycle states, however it cannot be excluded that G0/G1 will (more or less rapidly) progress into S/G2/M, i.e. slower cell cycle but not exactly dormant. Question is, how many of the red G0/G1 cells are also p27 positive? This information is not provided as the p27 staining seems to be done separately (and quantification is not shown) and only in one selected ER+ model. The authors should perform combined FUCCI/p27 analysis in several ER+ and ER- models (not only the selected one) to be able to make a proper statement on dormancy in ER+ vs ER- breast cancers.

(2) Statements on EMP are (as often occurs in EMP/EMT-related manuscripts) influenced by forced constitute knockdown or overexpression of epithelial or mesenchymal genes, rather than isolating and studying naturally occurring E vs M cells in a given model. This is of limited clinical relevance. Further, these types of experiments seem to be only done with one cell line (MCF7), not sufficient to assess the generality of the findings.

(3) From the experiments above, authors conclude that EMP (a.k.a. EMT) does not favor tumor progression or metastasis. This concept is not novel (eg. Fischer et al., Nature 2015; Padmanaban et. al, Nature 2019, etc.). Further, statements on EMP and dormancy are not supported by sufficient data at this stage. The authors should isolate naturally-occurring E vs M cells from various models, inject them (e.g. iv, to observe lung metastasis) and determine ratios of dormant vs non-dormant DTCs at the metastatic site in each population.

(4) Experiments with dox-induced E-cad overexpression are interesting, but key controls are missing. Dox-treated mice with a control vector are missing (to exclude effects of dox on cancer cell growth, as it often occurs) and again, are only done with one model. More models should be tested. Further, is this aspect specific to Ecad or induction of an epithelial state by other means

also results in re-awakening/increased proliferation?

(5) More generally, how do the authors reconcile their findings (in a nutshell, dormant DTCs are more mesenchymal, and re-awakening occurs when switching to a more epithelial phenotype) with (1) overwhelming clinical evidence showing epithelial DTCs at all possible stages in breast cancer and with (2) higher metastatic ability of TN models, generally thought to be less epithelial than ER+ ones? If their model is correct, the more mesenchymal, the more cells would remain dormant and not grow into metastasis. My worry is that statements on EMP and dormancy are too much extrapolated from the bigger picture, i.e. inherent features (EMP-independent) of cancer cells themselves, such as faster or slower proliferation rates, expression of metastasis promoting genes, interactivity with immune cells (in this case, restricted to the myeloid population) etc.

Reviewer #3 (Remarks to the Author):

The study by Aouad et al investigates the role of EMP programs to enable DTCs to acquire and eventually emerge from dormancy-associated phenotypes. Using the MIND inoculation model for multiple breast xenografts, the authors show that post-EMT cells are less proficient in underdoing metastatic outgrowth, an event that may reflect the coincident activation of metastatic dormancy. Moreover, the authors show that inducing MET in DTCs significantly enhances their metastatic outgrowth and relapse. The paper is rigorous and presents an abundance of evidence to support these phenomena, which are well-established and ingrained in the scientific literature. Less well established is the notion that post-EMT DTCs acquire dormancy-associated phenotypes. Unfortunately, this aspect of the study is underdeveloped with respect to (i) convincingly demonstrating the acquisition of dormancy-associated phenotypes in DTCs, and (ii) clearly showing that post-EMT DTCs exhibit enrichment of dormancy-associated markers that give way to proliferative markers as DTCs transition through MET. Additionally, the manuscript is devoid of any mechanistic insights into the signals operant in eliciting post-EMT/dormant DTCs to shed these phenotypes and reactivate proliferation programs. Thus, some attempt to elucidate the signal(s) and trigger(s) capable of promoting metastatic recurrence is warranted, as is providing some evidence linking this work to human tissues. Additional comments and concerns are presented below under "Specific Comments."

Specific Comments:

- 1) The primary weakness of this otherwise interesting study relates to the need to provide additional evidence of dormancy-associated phenotypes beyond p27 expression and Ki67 cells. Some attempts to incorporate measurements of NR2F1, DEC2, ratios of phospho-p38 MAPK:phospho-ERK1/2, etc. are warranted. Equally important, the authors need to strengthen the connections that EMT programs are in fact driving DTCs to become dormant, doing so by monitoring the appearance of the aforementioned markers both in 3D-cultures and in vivo. Likewise, similar enhanced rigor should be provided for DTCs induced to shed their dormant phenotypes upon enforced expression of Ecad. Without these additional controls and findings, the paper essentially reinforces established roles of EMT/MET cycles in driving dissemination and outgrowth.
- 2) The study lacks any evidence related to the nature of the signals operant in stimulating dormant DTCs to reactivate proliferative programs. Some attempts to mechanistically link the induction of MET to these events is warranted and will significantly enhance the overall significance of these findings.
- 3) The overall clinical significance would be greatly enhanced by inclusion of two sets of data. First, while one appreciates the impact of dormancy to ER+ disease relapse, it was somewhat disappointing that no attempts to assess the role of ESR1 mutations on EMP and dormancy-associated phenotypes were explored herein. This is an important clinical question as the development of endocrine resistance due to ESR1 mutations could contribute to the reactivation of proliferative programs and altered EMP status. Second, and along these lines, some attempts to validate these events/findings in human tissue samples is warranted.
- 4) Figure 1: Additional dormancy-associated markers should be provided.

5) Figure 2: The findings are descriptive and largely uninformative and can be removed from the paper – i.e., it is unsurprising that DTCs will adopt and display disparate morphologies in distinct organ sites and tissue microenvironments. Without some mechanistic insights into these events, it seems prudent to remove these analyses for future follow-up studies.

6) Figure 3: Immunofluorescent analyses should show CK8 staining with that of post-EMT cells. Additionally, one wonders whether when DTCs acquire/undergo EMP programs to assume dormant phenotypes. As such, similar staining of primary tumors, particularly their leading edges, for these markers is important, as is determining the status of these markers in CTCs.

7) Figures 4 & 5: Data supporting EMP program activation is strong and rigorous; however, evidence providing strong linkages of EMP programs to altered dormancy status is not well-developed and should be provided using either 3D-cultures or in vivo.

RESPONSE TO REVIEWER COMMENTS

In response to the reviewers' comments we have prepared a **major revision of** our manuscript entitled "Epithelial-mesenchymal plasticity determines ER+ breast cancer dormancy and reacquisition of an epithelial state drives awakening". We have performed **substantial additional work** and thank the reviewers for their constructive comments that have helped us improve the manuscript.

The majority of *in vivo* studies addressing the many roles of EMT in tumor progression are based on experiments with different GEMMs (Brabletz et al., 2021) and have yielded partially contradictory results with regards to EMT in mammary carcinogenesis. While a large body of evidence implies EMT in tumor invasion and metastatic progression (Aiello et al., 2018; Pastushenko et al., 2018; Rios et al., 2019; Simeonov et al., 2021) several studies show that an E-cadherin expression and an epithelial state are maintained during invasion and important for efficient metastatic seeding (Koch et al., 2020; Padmanaban et al., 2019). The various GEMM mammary carcinoma models, in particular the widely used MMTV polyoma Middle T-model with viral oncogene-driven tumorigenesis in the mammary epithelium, result in rapidly growing ER- tumors. While these models offer many experimental opportunities, caution is required when extrapolating finding to the human disease.

We demonstrate using the intraductal xenograft approach, which models different breast cancer subtypes, including hormone sensitive ones, with high resemblance to the clinical counterparts (Fiche et al., 2019; Richard et al., 2016; Sflomos et al., 2016) that the biology of metastasis and the role of EMT in tumor progression is different between the, so far understudied, luminal and the TN molecular subtypes.

These findings support the idea that some of the conflicting observations in the literature may relate to tumor type-specific biologies.

To demonstrate the link between EMT and dormancy more directly, we have performed single cell RNA sequencing (scRNAseq) on FACS-sorted lung DTCs from MCF-7 intraductal xenografts. We show that dormant DTCs have mesenchymal features, whereas proliferative DTCs have epithelial features. We provide direct evidence that <0.6 % of the primary tumor cells but 27% of lung DTCs are in an EMT-state.

Different lines of experiments now confirm the link between EMT and dormancy with use various methods to induce EMT (Reviewer #1 and #2):

More specifically, the link between EMT and dormancy has been confirmed by:

1. *TWIST*, *SNAIL*, and *ZEB2* overexpression in MCF-7 *in vivo*
2. TGF β treatment of MCF-7 cells *in vitro*
3. scRNAseq data corroborating the correlation between EMT and dormancy and expression of recently identified dormancy markers in the quiescent DTCs

We have checked if other processes contribute to the dormant phenotype (Reviewer #1 and #2) and now show additional EMT transcription factors halting tumor progression and markers associated with dormancy.

We have included additional cell models (Reviewer 2) and controls (Reviewer #2) and provide additional readouts to confirm the dormant phenotype (Reviewer #3) including functional *in vivo* evidence of EMP states.

Furthermore, we have identified cytokine signaling as a potential mechanism for E-cadherin upregulation and MET thereby linking microenvironment to awakening and show that pSTAT3 binding to the E-cadherin promoter region is increased upon IL6 treatment in ER+ BC cells.

Below, we provide a point-by-point response to the reviewers' comments **in red**.

In the manuscript text file we have marked all changes with track changes or color highlighting but we have also submitted a cleaned version for easier reading.

POINT-BY-POINT RESPONSE TO REVIEWER COMMENTS

We thank reviewer 1 for pointing to the clinical relevance of our work and appreciating that our *in vivo* study is carefully conducted. The individual points raised are addressed below.

Reviewer #1 (Remarks to the Author):

The manuscript entitled “Epithelial-mesenchymal plasticity determines estrogen receptor positive (ER+) breast cancer dormancy and reacquisition of an epithelial state drives awakening” provides comprehensive data from human ER+ and TN breast cancer cell lines and PDX cells injected into the mammary ducts of immune deficient mice to support the relevance of epithelial-mesenchymal plasticity in tumor dormancy and metastatic recurrence.

In this study, authors observed that ER+ disseminated tumor cells (DTCs) exhibit lower proliferative indices when compared to TN DTCs, which correlates with a delayed metastatic recurrence. In this model, metastatic dormancy is reversed by the induction of CDH1 expression in ER+ DTCs. Using breast tumor cell lines and PDX models in the study complement each other to emphasize its clinical relevance to the late recurrence that characterizes ER+ breast cancer patients. The *in vivo* characterization of tumor models used are carefully analysis with detailed quantification. However, several key issues should be addressed to provide further support to the conclusion the authors intend to draw.

1. A major conclusion that the authors intend to draw is that induction of EMP does not promote tumor cell dissemination in ER+ tumors. The key data presented to support this conclusion is Fig. 4 showing that knocking down CDH1 or expression of ZEB1 inhibited primary tumor invasion and tumor dissemination in the lung and the bone. In contradiction with this study, CDH1 downregulation or Zeb1 overexpression are usually associated with increased invasive capability in tumor cells. While the authors show a very limited marker changes associated EMP in Fig. S4, it is not convincing whether knocking down E-cad or expression of ZEB1 indeed activates a functional EMT program. Instead, knocking down CDH1 and overexpression of ZEB1 could lead to apoptosis and reduced cell proliferation in various cell types, which could explain the reduced primary tumor growth and the benign tumor morphology presented in tumors with CDH1 knockdown or Zeb1 overexpression. Further functional characterization of the effect of CDH1 knockdown or Zeb1 overexpression on cell proliferation, apoptosis, adherents junction integrity, cell migration and invasion should be carried out in culture. Furthermore, in Fig. 4C, E-cad staining signal is very weak and hard to see whether E-cad is on the adherents junctions. For both primary tumor with CDH1 knockdown or Zeb1 overexpression, apoptosis markers and additional EMP markers, including E-cadherin, beta-catenin, vimentin and desmosome markers should also be stained to evaluate the EMP status of these tumor cells.

Following the reviewer’s suggestions, we have further functionally characterized the effect of *CDH1* knockdown or *ZEB1* overexpression:

1. Re further *in vitro* characterization of the effect of *CDH1* knockdown or *ZEB1* overexpression on cell proliferation, we show here for the reviewer that following infection and drug selection, MCF-7:sh*CDH1* and MCF-7:*ZEB1* show viability comparable to the controls i.e. >96% but reduced cell proliferation.

Bar plot showing the percentage cell viability in MCF-7:Vector, sh*CDH1*, and *ZEB1* cells. Data represent mean ± SD from two biological replicates.

Bar plot showing the relative *MKI67* expression (Exp.) in MCF-7:control Vector, sh*CDH1*, and *ZEB1* cells. Data represent mean ± SD from 4 biological replicates.

2. *In vitro*: we demonstrate morphological changes reflecting a functional EMT: new micrographs illustrate the loss of epithelial morphology and the acquisition of discohesive growth in culture both functional consequences of EMT, induced by sh*CDH1* (New Supplementary Figure 5e) or for *ZEB1* overexpression (New Supplementary Figure 5i).

The text was amended accordingly, page 10 lines 275 “*In vitro*, control infected cells formed epithelial islands, whereas both MCF-7:sh*CDH1* and MCF-7:*ZEB1* cells lost the cobblestone morphology and showed discohesive growth (Supplementary Fig. 5d,e,h,i)”.

3. In addition to the changes in expression of *ZEB1*, *SNAI1*, *SNAI2*, *TWIST*, *VIMENTIN*, *CDH1,2*, and 3 markers of EMP previously shown in Supplementary Figure. 4, we now show that the fibrillar collagen, *COL3A1*, recently shown to be a dormancy marker (Di Martino et al., 2022) is increased upon *CDH1* down-modulation as well as *ZEB1* and *ZEB2* overexpression, and that *CXCL12* is increased upon *CDH1* down modulation and *ZEB1* overexpression (New Supplementary Figure. This indicates that the genetic manipulations elicit a functional EMT resulting in a dormant state of the tumor cells.
4. Most importantly, we now provide *in vivo* evidence for a functional EMP. The new Figure. 4f shows histological sections of MCF-7:shSCR vs MCF-7sh*CDH1* as well as MCF-7:Vector versus MCF-7:*ZEB1* (New Figure 4l) intraductal xenografts stained with picosirius red that reveal differences in the deposition of fibrillar collagens around *in situ* lesions. Image quantification (new Figures 4g and 4m) reveals 2 and 4.7-fold increase in sh*CDH1* and *ZEB1* xenografts, respectively.

“Picosirius red staining followed by image analysis revealed a 1.85 fold increase in fibrillar collagen deposition in MCF-7:sh*CDH1* xenografts in support of *CDH1* down modulation inducing a functional EMT *in vivo* (Kalluri and Neilson, 2003; Peng et al., 2017, Sflomos et al., 2021) (Fig. 4f,g).”
Lines 292-5, page 11

and

“As with sh*CDH1*, picosirius red staining showed a thin layer of fibrillar collagen around the ducts of MCF-7:Vector xenografts. In MCF-7:*ZEB1* xenografts, the fibrillar collagen deposits were increased 4.5-fold (Fig. 4l,m) providing evidence of a functional EMT *in vivo* (Peng et al., 2017).”
Lines 327-320, page 12.

Furthermore, we have improved the micrographs showing E-cadherin staining and are now showing an additional higher magnification to illustrate that E-cadherin can be detected at adherence junctions in new Fig. 4c.

To assess apoptosis *in vivo*, we stained sections from MCF-7 shSCR and sh*CDH1* as well as MCF-7 vector and MCF-7:*ZEB1* intraductal xenografts with anti-cleaved cytokeratin-18 antibody. We show that *CDH1* knockdown increases the percentage of cleaved-CK18+ cells from 0.6% in the control to 1.5%, $P < 0.001$ (New Supplementary Figure 6f). *ZEB1* overexpression had no significant difference (1.5% of positive cells in vector control cells vs 2.5% in *ZEB1* over expressors, $P = 0.11$) (New Supplementary Figure 6n). The data was added to Supplementary Figure 6, and text was amended in page 11, line 301-3 “The apoptotic index as determined by IF for cleaved CK18 increased from 0.6% in controls to 1.5% in MCF-7:sh*CDH1* xenografts ($P < 0.001$) (Supplementary Fig. 5f), and in page 12, line 330-1 “A trend for an increased apoptotic index was observed upon *ZEB1* overexpression (Supplementary Fig. 6n)”

Following the Reviewer's comment, we have further evaluated the EMT status of the primary tumors by IF using anti beta-catenin, p120 catenin, and vimentin antibodies. The results show that beta-catenin, E-cadherin, and p120 protein disappear from adherens junctions upon E-cad knockdown or Zeb1 overexpression, new Supplementary Figures 6d and 6o text lines 298-9 "CDH1 knockdown resulted in decreased protein levels of p120, and β -catenin at the adherens junctions as assessed by IF (Supplementary Fig. 6d)." and "Zeb1 overexpression decreased protein levels of E-cad, p120, and β -Catenin as assessed by IF and occasional Vim+ cells were detected (Supplementary Fig. 6o)", lines 331-3.

2. Since this conclusion that EMT inhibits tumor invasion and dissemination is contradictory to vast amount of published data demonstrating that induction of EMT promotes tumor cell invasion and dissemination into the circulation, additional approaches, such as TGF-beta treatment and induction of Snail1 or Twsit1 should be used to trigger EMT and test their impact on MCF7 tumor invasion and dissemination. The reason to do so is because that knocking CDH1 in epithelial tumor cells is not equivalent to activation of EMT since loss of CDH1 without concurrent activation of other mesenchymal program could lead to severe apoptosis. Similarly, Zeb1 activation alone may not be sufficient to induce EMT and might lead to apoptosis and growth inhibition. Therefore, additional approaches are needed to support this conclusion.

We thank the reviewer for this insightful comment. Indeed, EMT was originally observed in the tumor front of oesophageal/colorectal cancer and since then EMT has been shown to promote tumor invasion in many different models. However, it has also been reported that tumor cells require E-cadherin for metastasis (Padmanaban et al., 2019).

These apparently contradictory findings may relate to the different experimental systems and tumor types used. Indeed, we show that by the same experimental approach two different subtypes of human breast cancer, ER+ and TN, differentially regulate EMT markers and expose different metastatic behaviors. This argues that the role for EMT differs between specific tumor types and may, at least in part, account for the apparent contradictions within the literature and the discrepancies between our finding in the ER+ BC models and the vast GEMM literature, which represents ER- mammary carcinoma models.

As per reviewer's request, we have taken additional approaches:

1. We now provide evidence that ectopic expression of additional EMT transcription factors, *ZEB2*, *TWIST1*, and *SNAI1*, slows tumor growth and weight, as well as dissemination." Overexpression of other EMT-TFs such as *ZEB2*, which was consistently increased in lung DTCs (Fig. 3g), *TWIST-1*, and *SNAI1* all similarly reduced tumor growth and metastatic load of MCF-7 cells upon intraductal grafting (New Supplementary Figure. 6q,r). Taken together, these data indicate that different EMP states induced by distinct means in MCF-7 cells do not favor tumor progression nor metastatic propensity." Lines 336-340, page 13.
2. An *in vitro* treatment of MCF-7 with TGF- β *in vitro* resulted in decreased proliferation and ability to form colonies as shown in the plot below for the reviewer:

Treatment of MCF-7 with TGF- β *in vitro* reduces the colony formation ability and proliferation of the cells. **a.** Representative images of 3 independent replicates with or without TGF- β . **b.** Bar plot showing the number of colonies from 4 independent replicates. **c.** Relative absorbance (550 nm) after dissolving the crystals with methanol. Data represent mean \pm SD from 4 biological replicates.

3. Fig. 5 clearly demonstrated that DTCs of MCF7 tumors in distant sites present a more mesenchymal phenotype with endogenously induced EMT-TFs. But it is unclear when EMT is activated during MCF7 cell dissemination from primary tumors to distant sites. The conclusion that EMT didn't occur in primary tumors is indirectly deduced based on the data from CDH1 knockdown and Zeb1 overexpression. To support such conclusion, it is important to examine primary tumors to detect such rare tumor cells that have undergone EMT. Especially, Yu et al. (2013, Science) reported that even ER+ breast cancer patients have CTCs that present a mesenchymal molecular signature. Therefore, it is also important to examine circulating tumor cells from the blood in MCF7 tumor-bearing mice for EMP features.

We thank the reviewer for this comment and agree that it is not possible to conclude that EMT is induced at the distant site. We conclude merely that EMP at the primary site does not favor metastasis.

“Taken together, these data indicate that different EMP states induced by distinct means in MCF-7 cells do not favor tumor progression nor metastatic propensity.” Lines 335f.

The paper by Yu et al 2013 reports mesenchymal features of CTCs in breast cancer patients, some of whom had ER+ disease. This study, as most CTC studies, was conducted with metastatic breast cancer patients at advanced stage of the disease. At this stage, macro metastases are present while the primary tumor has typically been removed, often years earlier. Therefore, the CTCs are derived from metastatic lesions and do not provide any information about the cells which leave the primary tumor.

In the present study, we focus on ER+ xenograft models at an earlier stage: the tumor at the primary site is present, tumor cells have disseminated but no macro metastases are detected.

We have now performed single cell RNA sequencing on DTC and primary tumors and provide evidence that there are <1% primary tumor cells in an EMT state (New Figure 6a,c and New Supplementary Figure 9a). By the time DTCs become detectable in the blood drawn from mice with ER+ intraductal xenografts (> 3months), micro metastases are present and the site of origin of any given CTC is hence unclear.

Work from the K. Pantel and N. Aceto group suggest that CTCs retain their epithelial phenotype and that this even enhances their metastatic potential.

4. The authors determined that dormant ER+ DTCs presented mesenchymal features (Fig. 3) and inducing a transition to more epithelial phenotypes through E-cadherin overexpression promotes their growth in the secondary site (Fig. 5). Authors made opposite observations in TN DTCs, which rapidly progressed to macrometastatic lesions (Fig. 1). To further support their model, authors should determine whether rapid macrometastatic outgrowth detected in TN breast cancer cell lines is associated to the presence of epithelial phenotypes and mesenchymal-epithelial transition in the secondary site.

To address whether the rapid macro metastatic outgrowth detected in TN BC (BT20 and HCC1806) xenografts is associated with the presence of epithelial phenotypes and mesenchymal-epithelial transition in the secondary site, we now sacrificed xenografted mice earlier, that is 3 weeks after intraductal engraftment. At this stage, DTCs are detected in distant organs by luciferase activity but not large enough for detection by stereomicroscopy i.e. similar to the micro metastases we describe in the ER+ BC models (Figure 2a,c,f). Comparison of the expression levels of *MKI67*, *CDH1*, *EPCAM*, and EMT-TFs in these early (3 weeks) lungs versus lungs (harvested at 5-6 weeks after intraductal injection) with macro metastases, shows similar expression levels of epithelial and EMT-related genes (New Figure 3k,l). There is a trend of increasing epithelial gene expression in the lungs from HCC1806 xenografts (New Figure 3k,l).

The text was added at page 9, lines 241-9 “To determine whether TN DTCs pass through a dormant state prior to becoming macro-metastases, we compared the expression levels of epithelial and EMP-marker genes in the lungs at an earlier stage when micro-metastases prevail i.e. 3 weeks (Fig. 3k) versus 5-6 weeks after intraductal injection when macro-metastases are detected. At both time points, similar expression levels of *MKI67*, *CDH1*, *EPCAM*, and EMT-TFs were observed in lungs of mice engrafted with BT20 and HCC1806 cells (Fig. 3l).

Additionally, we examined the expression levels of *CDH1*, *MKI67* and EMT-TFs in primary tumors of the ER+ MCF-7 cells and the TN BT20, and HCC1806 cells. As shown in the New Supplementary Figure 4f TN BC cells show co-expression of *CDH1* and *SNAI1*, *SNAI2*, *VIM* at the primary site and proliferate faster than MCF-7

cells. The new description of the results can be found in pages 9-10, lines 249-254 “Comparison of marker gene expression in primary tumors of ER+ MCF-7 cells and TN BT20 and HCC1806 cells showed that highly proliferative TN BC cells co-express *CDH1* and *SNAI1*, *SNAI2*, *VIM*, and *TWIST-1* indicating that they are in a hybrid EMT state both in the primary tumor (Supplementary Fig. 4f) and in the DTCs. Thus, mesenchymal morphology and increased expression of multiple EMP markers are features specific to ER+ DTCs.”

As such EMT related genes are expressed at increased levels in TN vs ER+ BCs. Of note, while the mesenchymal phenotype associated with more aggressive disease course in TN BC we find anti correlation between EMT and cell proliferation and tumor progression in the ER+ BC models we study.

5. Suppl Fig. 5a-d shows that lung and brain-derived DTCs restore their proliferative ability when cultured in vitro. This enhanced proliferation in vitro correlates with changes in *CDH1* and *ZEB1* expression, in accordance with the hypothesis that reacquisition of epithelial phenotypes promotes cell proliferation. However, some of the EMT-TFs analyzed in fig. 5b show some inconsistency with this observation. Concretely, *SNAI2* expression is increased by 10-fold in Brain DTCs cultured in vitro, and *ZEB2* is increased by 5- and 50-fold in lung and brain DTCs, respectively. Since these changes are much bigger than the ones observed in *CDH1* and *ZEB1* expression, authors should evaluate the relevance of downregulating expression of each EMT-TF in to see which are more relevant for cell proliferation.

To address the reviewer’s concern about the non-uniform effect of the *ex vivo* culture on different EMT transcription factors, we show that brain DTCs have lower *CDH1* expression levels than lung DTCs *in vivo* (Figure 3f). This may explain that brain DTCs take longer than lung DTCs to emerge from the quiescent state. To test this hypothesis, we prolonged the 2-month observation period in culture to 4 months. Indeed, at this later time point *ZEB2* mRNA levels have decreased new Supplementary Fig. 4. Similarly, the expression levels of *SNAI2* drop 2-fold ($P=0.07$). We added micrographs showing brain DTCs *ex vivo* at similar timepoints (at 3 days and 2 months) in new Supplementary Figure. 7b, and text:

“The lung-derived DTCs gradually resumed proliferation after 2 months in culture and formed epithelial islets (Supplementary Fig. 7b). The brain DTCs, which had lower *CDH1* transcript levels than the lung DTCs (Fig. 3f) took longer to emerge from quiescence and formed epithelial islets at 4 months (Supplementary Fig. 7c). *CDH1* transcript levels were ultimately restored, and expression of the EMT-TFs, *ZEB1*, *ZEB2* and *VIM* decreased after serial passages in culture (Supplementary Fig. 7d-g), which is consistent with the hypothesis that reacquisition of an epithelial state enables cell proliferation.” Lines 362-9, page 14.

Technical points:

1. The authors state that MCF-7:shSCR cells invaded the stroma whereas the MCF-7:shCDH1 cells remained in situ, providing images of the two conditions in figure 4F. However, the image of MCF-7shSCR condition does not show the edge of the tumor nor stromal infiltration but only a tumor that is much bigger in size.

We thank the reviewer for this helpful comment and have now replaced the H&E of shSCR with new Figure 4e, which shows an area where tumor cells infiltrate the stroma.

2. Data represented in Figure 4D regarding the expression of androgen, estrogen and progesterone receptors requires more explanation in the main text.

More explanation about the expression of *ESR1*, *PgR*, and *AR* has been added to the text, “Down-modulation of E-cad decreased cell proliferation (Supplementary Fig. 5f) and increased *ZEB1* transcript levels (Supplementary Fig. 5g), but did not alter the expression of the hormone receptors *ESR1*, *PGR* nor the androgen receptor (*AR*) frequently expressed in ER+ BCs (Supplementary Fig. 5g). *ZEB1* overexpression decreased cell proliferation (Supplementary Fig. 5j), reduced *CDH1* and the transcript levels of *ESR1* and expression of downstream effector *PGR* at mRNA and protein levels (Zhang et al., 2017), and increased *CDH2* expression levels (Supplementary Fig. 5h,k).” Page 11, lines 277-283.

Reviewer #2 (Remarks to the Author):

Aouad and colleagues investigate mechanisms determining tumor cells dormancy and reawakening in mouse models using intraductal injection. They find a link between epithelial-mesenchymal plasticity (EMP) and induction of dormancy, and observe that E-Cadherin forced expression enables to overcome dormancy and promote metastasis growth. While the authors should be commended for their focus on this highly relevant topic (DTCs), altogether, this study suffers from several major weaknesses:

(1) The authors use different strategies to determine whether or not (and how many) DTCs are dormant (proliferative index, FUCCI, p27 staining). While Ki67/proliferative index mainly tells about proliferative rates and not necessarily about dormancy, FUCCI highlights cells at different cell cycle states, however it cannot be excluded that G0/G1 will (more or less rapidly) progress into S/G2/M, i.e. slower cell cycle but not exactly dormant. Question is, how many of the red G0/G1 cells are also p27 positive? This information is not provided as the p27 staining seems to be done separately (and quantification is not shown) and only in one selected ER+ model. The authors should perform combined FUCCI/p27 analysis in several ER+ and ER- models (not only the selected one) to be able to make a proper statement on dormancy in ER+ vs ER- breast cancers.

We thank the reviewer for the excellent suggestion and have now performed triple Co-IF staining with anti CK8, anti-Ki67 and anti-p27 antibodies on lungs from both ER- and ER+ xenograft models. As lung parenchymal cells express little to no CK8, this antibody identifies the disseminated BC cells. We show that the proportion of p27+/Ki67- CK8+ cells among the all CK8+ in TN DTCs is almost zero while it is between 10 to 20% in the ER+ models, MCF-7, T47D, and METS15 (New Figure 2m,n and New Supplementary Figure 3a,b). and text “To test for dormancy, we conducted a co-immunofluorescence (Co-IF) labeling for p27, a marker of dormancy (Bragado et al., 2013), Ki67, and Cytokeratin8 (CK8). The latter was used to distinguish the CK8+ DTCs because from the surrounding CK8^{LOW} mouse alveolar cells. This showed that less than 1% of the CK8+ cells in the lungs of mice engrafted with TN BC cells, BT20 and HCC1806 were p27+ and Ki67- (Fig. 2m,n and Supplementary Figure 3a). In contrast, p27+ Ki67- DTCs in lung sections from MCF-7, T47D, and METS15 bearing mice represented in 17.3, 13.7, and 22.5%, respectively, of the total CK8+ DTCs, (Fig. 2m,n and Supplementary Fig. 3b). Text was added at page 8, line 200-7.

(2) Statements on EMP are (as often occurs in EMP/EMT-related manuscripts) influenced by forced constitute knockdown or overexpression of epithelial or mesenchymal genes, rather than isolating and studying naturally occurring E vs M cells in a given model. This is of limited clinical relevance. Further, these types of experiments seem to be only done with one cell line (MCF7), not sufficient to assess the generality of the findings.

To address the reviewer’s concern:

1. We have now portrayed both primary tumor and lung DTCs by single cell sequencing. This reveals that 27% of MCF7 cells, which disseminated to the lungs have EMT features whereas <1% of the cells in the primary tumor do. New Figure 6 and Supplementary Fig. 9, the text was added accordingly, pages 16-18.
2. Furthermore, we measured the transcript levels of some of genes, like *CXCL12*, *COL3A1*, *BMP2*, *TGFBR2*, *FN1*, which we found upregulated in the most dormant subpopulation of DTCs in the lungs from mice engrafted with ER+ PDX METS15. We found them to be upregulated in the patient-derived DTCs, new Supplementary Fig. 9g. “ These findings extended to the lung DTCs of ER+ PDX METS15, in which *CXCL12*, *BMP2*, *TGFBR2*, *FN1* transcripts were upregulated compared to their respective primary tumor cells and *COL3A1* to a lesser extent (Supplementary Fig. 9g). Lines 471-3, page 17-18.
3. The functional EMT experiments with forced knockdown of *CDH1* or overexpression of *ZEB1* were not only performed with MCF-7 cells (Figure 4d-j) but also with two ER+ PDXs, T99 and METS15

(Figure 4o-t). The outcome of *CDH1* down modulation and *ZEB1* overexpression is similar to what we observed in MCF-7 xenografts suggesting that the observations are generalizable to ER+ BCs.

(3) From the experiments above, authors conclude that EMP (a.k.a. EMT) does not favor tumor progression or metastasis. This concept is not novel (eg. Fischer et al., Nature 2015; Padmanaban et. al, Nature 2019, etc.). Further, statements on EMP and dormancy are not supported by sufficient data at this stage. The authors should isolate naturally-occurring E vs M cells from various models, inject them (e.g. iv, to observe lung metastasis) and determine ratios of dormant vs non-dormant DTCs at the metastatic site in each population.

We thank the reviewer for raising this important point allowing us to clarify that our study specifically addresses the role of EMP in ER+ breast cancer. To our knowledge we study the endogenous metastatic process of human ER+ BC cells for the first time. The difficult to grow tumor cells are usually directly injected into the blood or the distant site.

To address the reviewer's concern, we have performed single cell sequencing analysis of cells disseminated to the lungs and primary tumor cells from the same host. This shows that the fraction of EMT cells is <1% at the primary site and up to 27% in the lungs.

At this point, the number of EMT cells from the primary tumor we are able to recover for the experiments the reviewer proposes are unfortunately too low. Based on the observations that EMP does not favor metastasis and that E-cad levels need to be tightly regulated we argue that it is unlikely these rare E-cad^{LOW} primary tumor cells seed metastases." Discussion, Line 561-ff .

(4) Experiments with dox-induced E-cad overexpression are interesting, but key controls are missing. Dox-treated mice with a control vector are missing (to exclude effects of dox on cancer cell growth, as it often occurs) and again, are only done with one model. More models should be tested. Further, is this aspect specific to Ecad or induction of an epithelial state by other means also results in re-awakening/increased proliferation?

We thank the reviewer for raising this important point. We now show that DOX has negligible effects on the body weight of mice (n=5 in each of the 3 independent cohorts). Importantly, the DOX treatment did not affect cell proliferation nor tumor progression or metastatic burden in the lungs. New Supplementary Figure 7j-m: "DOX treatment *per se* had negligible effects on the weight of mice (Supplementary Fig. 7j); it neither affected tumor growth nor the weight of the engrafted glands or the resulting micro-metastatic burden at endpoint (Supplementary Fig. 7k-m).." lines 390-3, page 15.

Following the reviewer's advice, we have now extended the inducible restoration of E-cadherin to another ER+ BC cell line model, the T47D cells. 5 months after intraductal injection, mice were randomized based on primary radiance, and fed either Dox-containing chow or normal chow for 2 months. T47D primary tumor growth was not significantly reduced New Supplementary Fig. 7n, o but radiance emanating from the lungs was increased by 3-fold New Supplementary Figure. 7p. Fluorescence stereomicrographs revealed larger lesions and image analysis showed a 10-fold increase in the area and intensity of lesions, upon E-cadherin restoration New Supplementary Figure 7q-s.

"To assess whether the findings in MCF-7 cells extended to other ER+ BC models, we generated T47D:*E-cad*^{IND} cells and followed the same experimental workflow (Fig. 5f). Primary tumor growth and mammary gland weight were not significantly affected by overexpression of E-cad (Supplementary Fig. 7n,o), but the metastatic load as assessed by ex vivo lung radiance increased 3-fold (Supplementary Fig. 7p). Analysis of fluorescence stereographs showed a 10-fold increase in area and fluorescence intensity of lesions upon E-cad restoration (Supplementary Fig. 7q-s). Thus, in two different ER+ BC models E-cad overexpression with resulting MET is sufficient to drive DTCs out of dormancy." Page 15-16, Lines 411-418.

(5) More generally, how do the authors reconcile their findings (in a nutshell, dormant DTCs are more mesenchymal, and re-awakening occurs when switching to a more epithelial phenotype) with (1) overwhelming clinical evidence showing epithelial DTCs at all possible stages in breast cancer and with (2) higher metastatic ability of TN models, generally thought to be less epithelial than ER+ ones? If their model

is correct, the more mesenchymal, the more cells would remain dormant and not grow into metastasis. My worry is that statements on EMP and dormancy are too much extrapolated from the bigger picture, i.e. inherent features (EMP-independent) of cancer cells themselves, such as faster or slower proliferation rates, expression of metastasis promoting genes, interactivity with immune cells (in this case, restricted to the myeloid population) etc.

We appreciate the reviewer's insightful concern and would like to clarify that our claim that EMT is linked to dormancy and a non-proliferative state is specific to ER+ breast cancers, more specifically ER+ non lobular breast cancers examined here. Indeed, throughout the manuscript we provide evidence that TN BC cells show very different behavior from ER+ BC cells. The anti-correlation of mesenchymal state and cell proliferation observed in ER+ BC cells does not apply to the TN BC cells.

We examined the expression levels of *CDH1*, *MKI67* and EMT-TFs in primary tumors of the ER+ MCF-7 cells and the TN BT20, and HCC1806 cells and show that the highly proliferative TN BC cells are in a hybrid EMT state (co-expression of *CDH1* and *SNAI1*, *SNAI2*, *VIM*) yet they proliferate faster than MCF-7, new Supplementary Figure 4f.

Reviewer #3 (Remarks to the Author):

The study by Aouad et al investigates the role of EMP programs to enable DTCs to acquire and eventually emerge from dormancy-associated phenotypes. Using the MIND inoculation model for multiple breast xenografts, the authors show that post-EMT cells are less proficient in underdoing metastatic outgrowth, an event that may reflect the coincident activation of metastatic dormancy. Moreover, the authors show that inducing MET in DTCs significantly enhances their metastatic outgrowth and relapse. The paper is rigorous and presents an abundance of evidence to support these phenomena, which are well-established and ingrained in the scientific literature. Less well established is the notion that post-EMT DTCs acquire dormancy-associated phenotypes. Unfortunately, this aspect of the study is underdeveloped with respect to (i) convincingly demonstrating the acquisition of dormancy-associated phenotypes in DTCs, and (ii) clearly showing that post-EMT DTCs exhibit enrichment of dormancy-associated markers that give way to proliferative markers as DTCs transition through MET. Additionally, the manuscript is devoid of any mechanistic insights into the signals operant in eliciting post-EMT/dormant DTCs to shed these phenotypes and reactivate proliferation programs. Thus, some attempt to elucidate the signal(s) and trigger(s) capable of promoting metastatic recurrence is warranted, as is providing some evidence linking this work to human tissues. Additional comments and concerns are presented below under "Specific Comments."

We thank the reviewer very much for the constructive comments. We now present additional data to

- (i) convincingly demonstrating the acquisition of dormancy-associated phenotypes in DTCs
- (ii) clearly show that post-EMT DTCs exhibit enrichment of dormancy-associated markers that give way to proliferative markers as DTCs transition through MET.
- (iii) provide potential mechanistic insights into the signals operant in eliciting post-EMT/dormant DTCs to shed these phenotypes and reactivate proliferation programs.

Specific Comments:

1) The primary weakness of this otherwise interesting study relates to the need to provide additional evidence of dormancy-associated phenotypes beyond p27 expression and Ki67 cells. Some attempts to incorporate measurements of NR2F1, DEC2, ratios of phospho-p38 MAPK:phospho-ERK1/2, etc. are warranted. Equally important, the authors need to strengthen the connections that EMT programs are in fact driving DTCs to become dormant, doing so by monitoring the appearance of the aforementioned markers both in 3D-cultures and in vivo. Likewise, similar enhanced rigor should be provided for DTCs induced to shed their dormant phenotypes upon enforced expression of Ecad. Without these additional

controls and findings, the paper essentially reinforces established roles of EMT/MET cycles in driving dissemination and outgrowth.

We thank the reviewer for this constructive comment. We have now performed single cell RNA sequencing on tumor cells isolated from the primary site and the lungs (New Figure 6 and New Supplementary Figure 9). The analysis reveals 4 clusters among the DTCs. Epithelial markers are correlated with increased DNA repair, proliferation markers, oxidative phosphorylation and MTOR signaling. Of note, among the two quiescent populations one is further enriched for EMT and is characterized with increased expression of ECM genes such as COL3A1, FN, recently implicated as dormancy-specific markers. Please refer to Pages 16-18 in text.

We have performed IF for both p-p38 and p-ERK across xenografts. While we could readily detect p-ERK in the nuclei of TN lung DTCs but not in the ER+ DTCs, we failed to get the p-p38 antibody to work. Of note, previous findings were in *in vitro* model and performed by immunoblotting (Aguirre-Ghiso, J. et al. 2003 Cancer Research), something we are unable to do in the present *in vivo* models because of the **small sample size**.

We additionally provide a quantification of p27+/Ki67- cells in multiple xenografts to strengthen our conclusion that dormancy is specific to the ER+ BC subtype in new Figure 2m,n and Supplementary Figure 3a,b.

2) The study lacks any evidence related to the nature of the signals operant in stimulating dormant DTCs to reactivate proliferative programs. Some attempts to mechanistically link the induction of MET to these events is warranted and will significantly enhance the overall significance of these findings.

We thank the reviewer for this very constructive suggestion. To identify pathways that may induce MET and awakening, we performed single cell RNA sequencing on DTCs which identified 4 different cell clusters. We find positive enrichment for epithelial markers and proliferative signatures in 2 of the clusters, called the “active” ones (clusters 0 and 2) and compared their gene expression profiles to the two clusters expressing mesenchymal markers and ZEB1, the “inactive” ones (clusters 1 and 3) New Figure 6h. We find that IL6/JAK/STAT3, and TNF α signaling via NF κ B are enriched in the active populations, among other pathways (New Supplementary Table 6). This suggests that cytokine signaling downstream of changes in the microenvironment act as a trigger for awakening and is in line with findings that IL-6/STAT3 signaling is important in disease progression in ER+ BC (Siersbæk et al., 2020). Analysis of ChIP Seq data from this study shows that Il-6 induces binding of p-STAT3 to the *CDH1* promoter both in MCF-7 and T47D cells consistent with E-cadherin being regulated by STAT3 activity. Furthermore, Il-6 induces binding of p-STAT3 as at ER promoter both in MCF-7 and T47D cells when the ER response signatures are increased in the more epithelial DTCs. Data was added to Figure 6 h, I and in the new Supplementary Figure 10. Text was incorporated accordingly in page 18 lines 479-492.

3) The overall clinical significance would be greatly enhanced by inclusion of two sets of data. First, while one appreciates the impact of dormancy to ER+ disease relapse, it was somewhat disappointing that no attempts to assess the role of ESR1 mutations on EMP and dormancy-associated phenotypes were explored herein. This is an important clinical question as the development of endocrine resistance due to ESR1 mutations could contribute to the reactivation of proliferative programs and altered EMP status. Second, and along these lines, some attempts to validate these events/findings in human tissue samples is warranted.

To address the reviewer’s comment about the *ESR1* mutations, we obtained MCF-7 cells harboring point mutations at the amino acids Y537 and D538 (MCF-7:Y537S and MCF-7:D538G, respectively) together with their parental cell lines (MCF-7:WT) from Dr. Simak Ali, Imperial College London. We engrafted them intraductally to the mammary glands of *NSG* female mice. The *in vivo* growth of MCF-7:Y537S was comparable ($P=0.6$) to MCF-7:WT, while MCF-7:D538G grew 2-fold faster than their WT counterpart ($P<0.0001$). The metastatic burden in the different organs was comparable between MCF-7:WT cells, and

both MCF-7:Y537S and MCF-7:D538G mutant cells. Thus, clinically relevant *ESR1* mutations are not sufficient to drive awakening. These data are shown in New Supplementary Figure 8a-c and text:

“*ESR1* mutations are not sufficient to lead to awakening

Our finding that MET is critical for awakening ER+ dormant DTCs begged the question as to what signals trigger this critical change. *ESR1* mutations occur frequently in patients treated with aromatase inhibitors and lead to metastatic recurrence (Jeselson et al., 2014). To test the hypothesis that *ESR1* mutations may cause metastatic awakening, we grafted MCF-7 cells, in which two hotspot mutations, Y537S and D538G were knocked into the endogenous *ESR1* locus (Harrod et al., 2017). The *ESR1*^{Y537S} allele did not affect primary tumor growth (Supplementary Fig. 7a) while MCF-7:*ESR1*^{D538G} cells proliferated faster compared to control MCF-7 cells (Supplementary Fig. 7b). In both cases, however, metastasis to different organs and the overall metastatic burden were not affected (Supplementary Fig. 7c) indicating that either of the two *ESR1* mutations is not sufficient to lead to awakening in the MCF7 MIND model.”

Lines 420-430.

Regarding the reviewer’s second point about validating our findings on human tissue samples, we selected some of the genes that were enriched in the dormant DTC population identified by scRNASeq, and validated them by qRT-PCR on matched primary and lung samples from the ER+ PDX METS15. We found *CXCL12*, *BMP2*, *TGFBR2*, *FN1* to be significantly upregulated in lung DTCs compared to their respective primary tumor cells; *COL3A1* to a lesser extent. The data are shown in the new Supplementary Figure 9g and described in the text: “These findings extended to the lung DTCs of ER+ PDX METS15, in which *CXCL12*, *BMP2*, *TGFBR2*, *FN1* transcripts were upregulated compared to their respective primary tumor cells and *COL3A1* to a lesser extent (Supplementary Fig. 9g). Lines 471-3, pages 17-18.

4) Figure 1: Additional dormancy-associated markers should be provided.

We have now performed Co-IF staining with anti CK8, anti-Ki67 and anti-p27 antibodies on lungs from both ER- and ER+ xenograft models.

As lungs are CK8 negative this antibody identifies the disseminated BC cells. We show that the proportion of p27+/Ki67- CK8+ cells among the all CK8+ in TN DTCs is almost zero while it is between 10 to 20% in the ER+ models, MCF-7, T47D, and METS15 new Fig. 3m,n and text: “Co-immunofluorescence (Co-IF) labeling for p27, a marker of dormancy (Bragado et al., 2013) Ki67, and Cytokeratin8 (CK8) to unequivocally identify DTCs because lung epithelial cells are CK8-. This showed that less than 1% of the CK8+ cells in the lungs of mice engrafted with TN BC cells, BT20 and HCC1806 were p27+ and Ki67-. In contrast, p27+ Ki67- DTCs in lung sections from MCF-7, T47D, and METS15 bearing mice represented in 17.3, 13.7, and 22.5%, respectively, of the total CK8+ DTCs, (Fig. 2m,n).” lines 201-206.

Furthermore, analysis of scRNAseq reveals a number of additional genes previously shown to characterize dormant DTCs, such as *CXCL12*, *BMP2*, *TGFBR2*, *FN1* as well as *COL3A1* significantly upregulated in dormant DTCs compared to active DTCs, New Supplementary Fig. 9f and described in the text: “The active cluster 2 contained some *MKI67*+ and *MCM2*+ cells while showing a negatively enriched EMT signature (Fig. 6g). The inactive cluster 3 showed increased expression of genes related to tumor-derived ECM and stromal crosstalk. These include *ELN*, *COL1A1*, *COL3A1*, *FN1*, *BMP2*, and *CXCL12* (Supplementary Fig. 9f), previously implicated in dormancy in different cancer models (Agarwal et al., 2019; Aguirre-Ghiso et al., 2003; Di Martino et al., 2022; Kobayashi et al., 2011; Montagner et al., 2020; Nobre, 2021). Together these findings suggest that DTCs have different degrees of EMP and dormancy, with cluster 3 representing the most dormant one.” Lines 463-470, Page 17.

5) Figure 2: The findings are descriptive and largely uninformative and can be removed from the paper – i.e., it is unsurprising that DTCs will adopt and display disparate morphologies in distinct organ sites and tissue microenvironments. Without some mechanistic insights into these events, it seems prudent to remove these analyses for future follow-up studies.

We thank the reviewer for this advice and have removed Figure 2 and Supplementary Fig. 2 from the manuscript.

6) Figure 3: Immunofluorescent analyses should show CK8 staining with that of post-EMT cells. Additionally, one wonders whether when DTCs acquire/undergo EMP programs to assume dormant phenotypes. As such, similar staining of primary tumors, particularly their leading edges, for these markers is important, as is determining the status of these markers in CTCs.

We have now performed co-IF of primary tumors with anti-CK8 and anti-E-cad antibodies. We show that CK8 and E-Cadherin double positive cells at the leading edge of tumors (i), in invasive areas or stroma (ii), and in cells adjacent to blood vessels (iii). New Figure 4c, and new Supplementary Figure 5a-c. Text: "IF showed that MCF-7 cells also retained E-cad and CK-8 protein expression observed in in situ and invasive lesions (Fig. 4c), as well as in areas of tumor budding (Supplementary Fig. 5a), at the leading invasive edge (Supplementary Fig. 5b), and in proximity to blood vessels (Supplementary Fig. 5c). Thus, EMP features are characteristic of ER+ DTCs and not readily detected by IF at the primary site." Lines 265-269, Page 10.

By this approach, we did not notice any cells undergoing EMT. As now revealed by sc RNASeq <1% of the tumor cells at the primary site have mesenchymal features and can hence be missed by IF.

To address the status of markers in whether there are rare cells that have undergone EMT, we performed scRNAseq on FACS-sorted primary MCF-7 tumour cells (n=3,200) 6 months after intraductal injection. We identified 8 distinct clusters, with 7 out of 8 expressing luminal epithelial genes namely, *ESR1*, *CDH1*, *KRT18*, and *EPCAM*, and only **one** mesenchymal (M)-like cluster (new Figure 7a), in which the EMT-related genes *ZEB2* and *VIM* were enriched. Gene set enrichment analyses revealed decreased enrichment scores for E2F and MYC targets, ER early and late response, glycolysis, and an increased enrichment score for EMT in M-like cluster 7 compared to the Epithelial-like clusters. Thus, M-like cluster 7 in primary tumour cells is in an EMT/dormant state. The percentage of primary tumour cells expressing EMT-TFs (*ZEB2*, *VIM*, and others) and low *CDH1*, is **0.59%**.

The analysis of CTCs is not informative with regards to the nature of cells leaving the primary tumor and giving rise to DTCs at distant sites. CTCs become detectable in the blood of xenografted animals only later (>3 months after MCF7 cell injection). At this stage, micro metastases are present in multiple organs and it cannot be determined whether any CTC isolated from the blood stems from the primary tumor or from distant organs.

7) Figures 4 & 5: Data supporting EMP program activation is strong and rigorous; however, evidence providing strong linkages of EMP programs to altered dormancy status is not well-developed and should be provided using either 3D-cultures or in vivo.

To address this reviewer's concern, we now provide *in vivo* data and show by single cell sequencing of MCF-7 cells from the primary tumor and from the lungs. Both the data from the primary tumor and the lung DTCs show a strong anti-correlation between epithelial gene signatures and dormancy, new Figure 6 and supplementary Figure 9. Cell proliferation, DNA damage response, oxidative phosphorylation, and mTOR signatures are positively enriched in the epithelial versus the more mesenchymal populations. (Supplementary Figure 9 and Table 6).

Furthermore, we analyzed expression of dormancy-implicated markers, such as *CXCL12*, *COL3A1*, and *FN1* identified by the scRNASeq analysis in lungs from mice engrafted with MCF-7:RFP-Luc. We show that knockdown of *CDH1* or *ZEB1* overexpression increases the transcript levels of *COL3A1* and *CXCL12*, and that *ZEB2* overexpression increases *FN1* transcript levels. These new data were added to Supplementary Figure 9h, and in text at page 18, lines 473-8.

"To test whether the experimentally induced EMP induced dormancy-related genes, we assessed the expression of *COL3A1*, *CXCL12*, and *FN1* in MCF-7:sh*CDH1*, :*ZEB1*, and :*ZEB2*. Both *CDH1* knockdown and *ZEB1* overexpression upregulated *COL3A1* and *CXCL12*, while *ZEB2* overexpression upregulated *FN1* (Supplementary Fig. 9h). Thus, EMT-TFs can induce a dormant state via the upregulation of dormancy-specific genes."

References:

- Aceto, N., Bardia, A., Miyamoto, D.T., Donaldson, M.C., Wittner, B.S., Spencer, J.A., Yu, M., Pely, A., Engstrom, A., Zhu, H., et al. (2014). Circulating tumor cell clusters are oligoclonal precursors of breast cancer metastasis. *Cell* 158, 1110–1122.
- Aiello, N.M., Maddipati, R., Norgard, R.J., Balli, D., Li, J., Yuan, S., Yamazoe, T., Black, T., Sahmoud, A., Furth, E.E., et al. (2018). EMT Subtype Influences Epithelial Plasticity and Mode of Cell Migration. *Dev. Cell* 45, 681-695.e4.
- Brabletz, S., Schuhwerk, H., Brabletz, T., and Stemmler, M.P. (2021). Dynamic EMT: a multi-tool for tumor progression. *EMBO J.* 40, e108647.
- Bragado, P., Estrada, Y., Parikh, F., Krause, S., Capobianco, C., Farina, H.G., Schewe, D.M., and Aguirre-Ghiso, J.A. (2013). TGF β 2 dictates disseminated tumour cell fate in target organs through TGF β -RIII and p38 α / β signalling. *Nat. Cell Biol.* 15, 1351–1361.
- Di Martino, J.S., Nobre, A.R., Mondal, C., Taha, I., Farias, E.F., Fertig, E.J., Naba, A., Aguirre-Ghiso, J.A., and Bravo-Cordero, J.J. (2022). A tumor-derived type III collagen-rich ECM niche regulates tumor cell dormancy. *Nat. Cancer* 3, 90–107.
- Fiche, M., Scabia, V., Aouad, P., Battista, L., Treboux, A., Stravodimou, A., Zaman, K., RLS, Dormoy, V., and Ayyanan, A. (2019). Intraductal patient-derived xenografts of estrogen receptor α -positive breast cancer recapitulate the histopathological spectrum and metastatic potential of human lesions. *J. Pathol.* 247, 287–292.
- Harrod, A., Fulton, J., Nguyen, V.T.M., Periyasamy, M., Ramos-Garcia, L., Lai, C.-F., Metodiev, G., de Giorgio, A., Williams, R.L., Santos, D.B., et al. (2017). Genomic modelling of the ESR1 Y537S mutation for evaluating function and new therapeutic approaches for metastatic breast cancer. *Oncogene* 36, 2286–2296.
- Jeselsohn, R., Yelensky, R., Buchwalter, G., Frampton, G., Meric-Bernstam, F., Gonzalez-Angulo, A.M., Ferrer-Lozano, J., Perez-Fidalgo, J.A., Cristofanilli, M., Gomez, H., et al. (2014). Emergence of constitutively active estrogen receptor- α mutations in pretreated advanced estrogen receptor-positive breast cancer. *Clin Cancer Res* 20, 1757–1767.
- Kalluri, R., and Neilson, E.G. (2003). Epithelial-mesenchymal transition and its implications for fibrosis. *J. Clin. Invest.* 112, 1776–1784.
- Koch, C., Kuske, A., Joosse, S.A., Yigit, G., Sflomos, G., Thaler, S., Smit, D.J., Werner, S., Borgmann, K., Gärtner, S., et al. (2020). Characterization of circulating breast cancer cells with tumorigenic and metastatic capacity. *EMBO Mol. Med.* 12, e11908.
- Padmanaban, V., Krol, I., Suhail, Y., Szczerba, B.M., Aceto, N., Bader, J.S., and Ewald, A.J. (2019). E-cadherin is required for metastasis in multiple models of breast cancer. *Nature* 573, 439–444.
- Pastushenko, I., Brisebarre, A., Sifrim, A., Fioramonti, M., Revenco, T., Boumahdi, S., Van Keymeulen, A., Brown, D., Moers, V., Lemaire, S., et al. (2018). Identification of the tumour transition states occurring during EMT. *Nature* 556, 463–468.
- Peng, D.H., Ungewiss, C., Tong, P., Byers, L.A., Wang, J., Canales, J.R., Villalobos, P.A., Uraoka, N., Mino, B., Behrens, C., et al. (2017). ZEB1 induces LOXL2-mediated collagen stabilization and deposition in the extracellular matrix to drive lung cancer invasion and metastasis. *Oncogene* 36, 1925–1938.
- Richard, E., Grellety, T., Velasco, V., MacGrogan, G., Bonnefoi, H., and Iggo, R. (2016). The mammary ducts create a favourable microenvironment for xenografting of luminal and molecular apocrine breast tumours. *J Pathol.*
- Rios, A.C., Capaldo, B.D., Vaillant, F., Pal, B., van Ineveld, R., Dawson, C.A., Chen, Y., Nolan, E., Fu, N.Y., 3DTCLSM Group, et al. (2019). Intraclonal Plasticity in Mammary Tumors Revealed through Large-Scale Single-Cell Resolution 3D Imaging. *Cancer Cell* 35, 618-632.e6.
- Sflomos, G., Dormoy, V., Metsalu, T., Jeitziner, R., Battista, L., Scabia, V., Raffoul, W., Delaloye, J.F., Treboux, A., Fiche, M., et al. (2016). A Preclinical Model for ER α -Positive Breast Cancer Points to the Epithelial Microenvironment as Determinant of Luminal Phenotype and Hormone Response. *Cancer Cell* 29, 407–422.
- Sflomos, G., Battista, L., Aouad, P., De Martino, F., Scabia, V., Stravodimou, A., Ayyanan, A., Ifticene-Treboux, A., Rls, Bucher, P., et al. (2021). Intraductal xenografts show lobular carcinoma cells rely on their own extracellular matrix and LOXL1. *EMBO Mol. Med.* 13, e13180.

Siersbæk, R., Scabia, V., Nagarajan, S., Chernukhin, I., Papachristou, E.K., Broome, R., Johnston, S.J., Joosten, S.E.P., Green, A.R., Kumar, S., et al. (2020). IL6/STAT3 Signaling Hijacks Estrogen Receptor α Enhancers to Drive Breast Cancer Metastasis. *Cancer Cell* 38, 412-423.e9.

Simeonov, K.P., Byrns, C.N., Clark, M.L., Norgard, R.J., Martin, B., Stanger, B.Z., Shendure, J., McKenna, A., and Lengner, C.J. (2021). Single-cell lineage tracing of metastatic cancer reveals selection of hybrid EMT states. *Cancer Cell* 0.

Reviewers' comments:

Reviewer #1 (Remarks to the Author):

In this revision, the authors have performed a number of experiments and addressed many issues raised successfully. However, the conclusion that EMP in primary tumors does not favor tumor progression nor metastasis is not supported by several new pieces of data presented. Therefore, my suggestion is that this conclusion should not be included in the manuscript due to insufficient data support, while the rest of the study is well revised for publication. The reasons are as follows.

1. Fig. 4b shows that Twist1 and Snai2 levels increased 15.6 and 8.6-fold at the invasive stage. But the text states that that EMP features are not readily detected at the primary site, which is not consistent with the data. These data could equally be explained as EMP is activated at the primary site, but the limited markers used (CDH1, ZEB1, and VIM) for IF did not capture such partial activation of EMP.
2. Several figures show that deletion of CDH1 or overexpression of ZEB1 or Snai1 or TWIST-1 decrease proliferation significantly. Because the way micrometastasis is detected using 100um as the cutoff for micro- vs. macro lesions on stereoscope cannot differentiate single DTCs vs. tiny (<10 cells) cluster of DTCs that have already proliferated several times, it is equally possible that the detected reduction of "micro-metastatic" load is due to reduced proliferation and higher apoptosis. Therefore, the conclusion that the difference in micrometastases is not due to proliferation is not sufficiently supported by the data.
3. While CTC analysis was suggested to help to resolve this issue, no such analysis is provided in the revision due to various reasons raised by the authors. Even if the CTCs could be due to disseminated cells from metastases, such data would still reveal whether CTCs present EMP characteristics since distant macrometastases are E-cad+EMT-TFlow again. Without such crucial data, the conclusion that EMT in the primary site does not favor metastasis remains lacking sufficient support.
4. Conceptually, Fig. 7 describes that E-cad+ EMT-TFlowKi67+ tumor cells disseminate into distant sites, where they become E-cadlowEMT-TFlowhighKi67low to enter dormancy. Tumor cells are selected constantly during development for traits to provide them the fitness to succeed in growth and dissemination. The issue is why cells undergo EMT if such conversion does not provide any advantage to tumor cells during metastasis, instead EMP inhibits metastasis. All the data presented in this study could equally support the conclusion that activation of EMP first occurs in primary sites in rare tumor cells to allow dissemination in these ER+ tumors; reversion to the epithelial states weakens dormant cells to grow into macrometastases.

Reviewer #2 (Remarks to the Author):

The authors properly answered some of the initial questions in their revised submission, however, some substantial weaknesses related to previous questions remain.

(1) The authors failed to conclusively demonstrate that naturally occurring E versus M cells (without any forced overexpression system) have differential dormancy at the DTC stage in ER+ breast cancer – which to my understanding, is the core message of the paper. The scRNAseq experiment is obviously descriptive, and not sufficient to conclusively demonstrate the above. For their statements to be supported, it would be beneficial to isolate naturally occurring E vs M cells from models where these are sufficient in number, and inject them directly to measure dormant vs non-dormant DTCs.

(2) Related to the new scRNAseq analysis: with a 10x approach, typically characterized by (very) shallow sequencing depth and massive gene dropout rates, it is challenging (dangerous, potentially misleading) to pinpoint expression levels of individual genes. As control, could they show RFP expression levels aside the E and M genes. Also, can they exclude that cells with higher expression of M genes (proportionally to E genes) are those with lowest quality, eg. looking at the number of detected features and percent of mitochondrial vs nuclear genes.

Reviewer #3 (Remarks to the Author):

The authors have done a masterful job of addressing my previous concerns, as well as those of the other reviewers.

We thank the reviewers for their careful reassessment, positive and constructive comments.

As asked by **reviewer 1**, we removed the conclusion that «EMP in primary tumors does not favor tumor progression nor metastasis” to address his/her remaining concern.

Reviewer 2 asked to demonstrate “that naturally occurring E versus M cells (without any forced overexpression system) have differential dormancy at the DTC stage in ER+ breast cancer, which to my understanding, is the core message of the paper.”

EMP (M) cells at the primary site constitute a rare population (0.6% in total). Unfortunately, we are unable to do this due to technical restraints. To conclude on metastatic propensity, we would require a minimum number of 7 mice injected with at least 2 mammary glands (total of 3,000,000 cells approximately). Therefore, we would need to FACSsort **600,000,000** primary tumor cells, which would take over 8 hours, beyond the working of our flow cytometry facility.

The reviewer made the excellent suggestion to isolate E and M cells “from models where they are sufficient and inject them”. Unfortunately we are unable to isolate sufficient numbers of ‘M’ cells from ER+ clinically relevant models and therefore unable to perform this experiment. The use of a different clinically not so relevant model would take away the main message of the paper namely studying ER+ BC cells in clinically relevant settings (with very slow growth and major clinical challenge: metastatic dormancy).

Moreover, while we agree that the presence of E and M cells is a very intriguing observation that warrants further investigation, this is not the core message of the manuscript (please see significance).

In addition to the functional genetic evidence that forced E-cadherin expression in DTCs leading to re-acquisition of an epithelial state drives metastatic proliferation, we now provide direct evidence that lung DTCs spontaneously transit from a M to an E state using RNA velocity, new supplementary Figure 10. To the best of our knowledge, this is the first time a dynamic state in a clinically-relevant model for ER+ BC is reported.

We have addressed the reviewer’s concern about the RNA seq data are addressed in detail below.

Reviewers' comments:

Reviewer #1 (Remarks to the Author):

In this revision, the authors have performed a number of experiments and addressed many issues raised successfully. However, the conclusion that EMP in primary tumors does not favor tumor progression nor metastasis is not supported by several new pieces of data presented. Therefore, my suggestion is that this conclusion should not be included in the manuscript due to insufficient data support, while the rest of the study is well revised for publication. The reasons are as follows.

We thank the reviewer for stating that we addressed many of the issues successfully for publication with the 58 data panels, which were added.

We have addressed the remaining concern and removed the conclusion that “EMP in primary tumors does not favor tumor progression nor metastasis”.

It was completely removed from the abstract and we have specified in the result section that our findings relate specifically in ER+ BC, line 338. For the remainder of the text it is always specified, which particular model the conclusions pertain to.

The intraductal models that we pioneered and use here are the first to reflect ER expression as in the clinical counterpart. As we point out in the **discussion there are clinical data to support that EMT while clearly correlating with tumor progression in some tumors (here head and neck) is unrelated to clinical factors in ER+ BC:**

“Using scRNAseq in head and neck carcinoma, the group of Itay Tirosh has defined a partial (p)-EMT signature, and found that it correlates with lymph node metastasis, N stage, reduced survival, and therapy resistance (Puram et al., 2018; Tirosh et al., 2016). Interrogating a similar signature gene-set in other cancer types (of interest, Luminal A and B breast carcinoma) revealed no correlation with any clinical factors (Tyler and Tirosh, 2021), suggesting that p-EMT is likely to exist in many contexts, both human and mice models, but that the exact signature/genes as well as its strength, frequency and role-in-metastasis are quite variable, and even minimal in HR+ BC.”

We have further addressed the reviewer’s individual points for clarification:

1. Fig. 4b shows that Twist1 and Snai2 levels increased 15.6 and 8.6-fold at the invasive stage. But the text states that that EMP features are not readily detected at the primary site, which is not consistent with the data. These data could equally be explained as EMP is activated at the primary site, but the limited markers used (CDH1, ZEB1, and VIM) for IF did not capture such partial activation of EMP.

We share the reviewer’s view that the limited number of markers used to test for EMP when we compared the *in situ* to the invasive stage (Fig. 4b) does not allow to exclude a partial activation of EMP. We have changed the text to indicate that we only used the features of EMP, which we readily detected in the DTCs in the same model. Lines 271f.

We would like to point out that the findings with a limited number of markers are matched with what we find in the sc RNA seq approach which includes many more features Fig.6. An EMP signature is readily detected in a large subpopulation of DTCs but less than 0.6% of the primary tumor cells.

The reported relative increase in *SNAI2* and *TWIST1* transcripts in the invasive versus *in situ* disease is not necessarily indicative of an EMP, as we previously detected it in the context of global gene expression profiling linked to a basal differentiation program elicited in MCF7 cells by the stromal versus intraductal environment, see discussion:

“The upregulation of *SNAI2* and *TWIST1* transcripts in the invasive versus *in situ* disease, is in line with our previous findings that compared to the intraductal xenograft approach the fat pad microenvironment induces these factors as **part of a basal differentiation program** triggered by TGFβ-signaling (Sflomos et al., 2016). As such, their elevated expression is likely a consequence of invasion and not causally related to it. In line with cell type-specific EMP states (Pastushenko et al., 2018), our findings suggest that in ER+ human BC cells, the critical EMT-TFs are *ZEB1/ZEB2* whereas *SNAI2* and *TWIST1*, critical for EMP in other tissues, control a distinct cellular differentiation program that is linked to basal features.”

2. Several figures show that deletion of CDH1 or overexpression of ZEB1 or Snai1 or TWIST-1 decrease proliferation significantly. Because the way micrometastasis is detected using 100um as the cutoff for micro- vs. macro lesions on stereoscope cannot differentiate single DTCs vs. tiny (<10 cells) cluster of DTCs that have already proliferated several times, it is equally possible that the detected reduction of

“micro-metastatic” load is due to reduced proliferation and higher apoptosis. Therefore, the conclusion that the difference in micrometastases is not due to proliferation is not sufficiently supported by the data.

While EMT acquisition decreases the proliferation of ER+ BC cells, it does not enhance their metastatic capacity. Moreover, there is evidence in the literature and from our own work that, at least in some tumor types, dissemination happens very early on (Sflomos, G. et al. 2016; Fiche, M. et al. 2019. Klein, C. 2009, 2020). We would like to stress the following findings:

1. Overexpression of ZEB2, TWIST1, or SNAI1 decreased cell proliferation approx. 1.5-fold. However, the metastatic burden was decreased 7.5-fold on average. Therefore, the cells that remained in the ducts and survived, failed to successfully seed metastasis in distant organs. In the first round of revision, the reviewer requested: “additional approaches, such as TGF-beta treatment and induction of Snail1 or Twisit1 should be used to trigger EMT and test their impact on MCF7 tumor invasion and dissemination.” We did this and also validated the presence of EMT features by multiple approaches.

2. To exclude the possibility that the decreased metastatic burden merely reflected decreased primary tumor growth, we performed a paired analysis of mice with similar tumor burden at endpoint. In these mice, the micro-metastatic load in the brain, lungs, liver, and bones was reduced by 90% (Suppl Figure 6h).

3. Overexpression of E-cadherin via an inducible approach decreased the proliferation of MCF-7 at the primary tumor site but increased their proliferation at distant sites (Figure 5). If the hypothesis that lower tumor burden equals lower metastasis, this is not applicable to this ER+ BC model.

4. Finally, we have gathered data from experiments, in which we surgically removed the mammary glands 4 months after engraftment with MCF7 cells. Neither the weight of mammary glands nor their radiance at the time of removal correlate with the metastatic burden detected later on (lung radiance in this case). This further suggests that many metastatic events occur before tumors become palpable.

Fig 1. **A.** Correlation plot between lung radiance and the weight of mammary glands, or **B.** sum of primary radiance post mastectomy. MCF-7 RFP-Luc were injected in the 4th mammary glands, and were grown for 4 months after which mammary glands were resected.

5. Our single cell data suggest the EMT cells found in the lungs of MCF-7 bearing mice, have reduced cyclin D1, Ki67, MCM2 expression. Interestingly, we did not see any increase in the apoptotic gene signature in those cells, and of interest, the 2 EMT cell populations had the lowest enrichment score for DNA Damage hallmark. On the contrary, EMT cells (clusters 1 and 3) have reduced apoptosis and DNA repair signatures, in line with their increased survival advantage and dormancy phenotype.

Fig. 2. Enrichment scores for 4 clusters in the scRNA seq of the lung DTCs from MCF-7 bearing mice.

Therefore, the inactive DTCs in the lungs have reduced proliferation, lower apoptosis rate, but higher EMT scores.

Taken together, at least in the ER+ BC models tested here, decreased primary tumor burden cannot be equaled to decreased metastatic burden.

PS:

The cut-off by 100 um the reviewer refers to was used only in the context of the IHC analysis of the TN BC MIND model (Figure 2h). Throughout the manuscript the term “micro metastatic load” is used for lesions detected by luminescence (IVIS Spectrum) measurements, which are not detected by eye.

3. While CTC analysis was suggested to help to resolve this issue, no such analysis is provided in the revision due to various reasons raised by the authors. Even if the CTCs could be due to disseminated cells from metastases, such data would still reveal whether CTCs present EMP characteristics since distant macrometastases are E-cad+EMT-TFlow again. Without such crucial data, the conclusion that EMT in the primary site does not favor metastasis remains lacking sufficient support.

Regarding both point 3&4, we would like to highlight that, unlike widely used GEMM models which give rise to rapidly growing tumors, in the intraductal ER+ BC models we describe here tumor growth is slow and even in host mice, which we kept over a year no macro metastases were detected. This reflects the clinical situation much more closely than any of the preexisting models in which there is a constant strong selection pressure on the rapidly growing tumor cells.

We have also addressed this point in the discussion:

Discussion: “There is also evidence, however, that an epithelial phenotype is important for tumor cell dissemination, CTC survival, and for metastatic outgrowth (Koch et al., 2020; Padmanaban et al., 2019). These apparent contradictions are partly resolved with the present study, in which we compare two different subtypes of breast cancer by the same experimental approach and show that the role of EMP as well as the biological properties of DTCs is tumor subtype-dependent. “

4. Conceptually, Fig. 7 describes that E-cad+ EMT-TFlowKi67+ tumor cells disseminate into distant sites, where they become E-cadlowEMT-TFlowhighKi67low to enter dormancy. Tumor cells are selected constantly during development for traits to provide them the fitness to succeed in growth and dissemination. The issue is why cells undergo EMT if such conversion does not provide any advantage to tumor cells during metastasis, instead EMP inhibits metastasis. All the data presented in this study could equally support the conclusion that activation of EMP first occurs in primary sites in rare tumor cells to allow dissemination in these ER+ tumors; reversion to the epithelial states weakens dormant cells to grow into macrometastases.

The rare EMT-like cells that we detected in the primary tumor by scRNAseq have low expression of *CDH1*. Our functional data **confirm** that E-cadherin is essential and required for metastasis. Therefore, it is unlikely that this small population of primary tumor cells (0.6%) gives rise to 35% EMT-like cells in distant sites simply because of the kinetics of these tumor cells.

Reviewer #2 (Remarks to the Author):

The authors properly answered some of the initial questions in their revised submission, however, some substantial weaknesses related to previous questions remain.

(1) The authors failed to conclusively demonstrate that naturally occurring E versus M cells (without any forced overexpression system) have differential dormancy at the DTC stage in ER+ breast cancer – which to my understanding, is the core message of the paper.

The scRNAseq experiment is obviously descriptive, and not sufficient to conclusively demonstrate the above. For their statements to be supported, it would be beneficial to isolate naturally occurring E vs M cells from models where these are sufficient in number, and inject them directly to measure dormant vs non-dormant DTCs.

We thank the reviewer for acknowledging the points we addressed in our revised manuscript. The core message of our work is that ER+ versus ER- BC cells have very distinct metastatic biologies and specifically ER+ DTCs bear EMT features. (The “differential dormancy at the DTC stage” is a finding we elaborated on in response to reviewers’ comments.)

The reviewer suggested a very nice experiment to address directly the nature of E and M cells. In the ER+ BC models we are using in this study, EMP (M) cells at the primary site constitute a rare population (0.6% in total). Unfortunately, we are unable to do this due to technical restraints. To conclude on metastatic propensity, we would require a minimum number of 7 mice injected with at least 2 mammary glands (total of 3,000,000 cells approximately). Therefore, we would need to FACS sort **600,000,000** primary tumor cells, which would take over 8 hours, beyond the working of our flow cytometry facility.

Moving to models where E and M cells are “sufficient in number, and inject them directly to measure dormant vs non-dormant DTCs” would demean the main message of this paper which is to examine mechanisms underlying dormancy in ER+ BC in a clinically relevant context.

To address the reviewer’s concern with the present ER+ intraductal in vivo models, we have now analyzed RNA velocity in the scRNAseq data set new Supplementary Figure 10:

"To test the hypothesis that mesenchymal-like DTCs transit to an epithelial state to drive recurrence, we assessed dynamic changes in mRNA expression based on spliced versus unspliced transcripts: RNA velocity (Bergen et al., 2020; La Manno et al., 2018). At the primary site the directionality of cells projected in the UMAP represented by velocity vectors varies between different subpopulations with the M cluster (7) not showing any directionality (Supplementary Fig. 10a). In the lung DTCs, the velocity vectors, specifically those at the bridge between inactive and active states, are directed towards the epithelial clusters in line with MET (Supplementary Fig. 10b). The phase plots of the epithelial genes, *KRT8*, *CDH1*, *ESR1*, and *EPCAM* indicate transcript induction whereas the phase plots of the mesenchymal genes, *VIM*, *S100AF*, and *TCF4* are consistent with repression of these transcripts (Supplementary Fig. 10c) thus supporting the hypothesis that ER+ BC seldom undergo EMP at the primary site and that dormant DTCs transit from a mesenchymal to an epithelial state." Page 18 Line 499-510

Fig. 3. New Supplementary Figure 10. RNA velocity points to MET in lung DTCs.

a, b. UMAP showing the steady-state RNA velocity arrows in MCF-7 cells at the primary site (a) and in the lungs (b) of intraductally xenografted mice. **c, d.** Representative phase plot showing spliced *versus* unspliced RNA. The linear line of slope represents the steady-state. The circular clockwise arc depicts the changes in spliced *versus* unspliced RNA in a cell at a given time. Phase plots of the epithelial genes, *KRT8*, *KRT18*, *ESR1*, *CDH1*, and *EPCAM*, and **d.** the mesenchymal genes, *VIM*, *TCF4*, and *S100A4*.

While we do not show any evidence which defines where and when the EMP occurs, in the discussion we refer to work from the Sahai lab that argues for a EMT at the distant site:

“It has been shown that EMP can be induced at the distant sites in a syngeneic mouse mammary tumor cell line model, which remains latent in the lungs following intravenous injection (Montagner et al.,

2020). In co-culture models, cross-talk between these mammary carcinoma cells and alveolar type-1 cells resulted in the induction of EMT-TFs (Montagner et al., 2020). “

(2) Related to the new scRNAseq analysis: with a 10x approach, typically characterized by (very) shallow sequencing depth and massive gene dropout rates, it is challenging (dangerous, potentially misleading) to pinpoint expression levels of individual genes. As control, could they show RFP expression levels aside the E and M genes. Also, can they exclude that cells with higher expression of M genes (proportionally to E genes) are those with lowest quality, e.g. looking at the number of detected features and percent of mitochondrial vs nuclear genes.

We thank the reviewer for raising the concern on challenges related to using 10X Genomics analyses. While sequencing depth is not as deep as other, very expensive approaches, it has become the standard approach in many fields and is widely used and accepted.

Importantly, the cells analyzed have at least 1,000 RNA features. This is the standard used for any single cell RNA analysis.

Fig. 4. UMAP showing the RNA features in lung DTCs.

To exclude the possibility that our results are biased by low quality cells, the analyses were restricted to cells characterized by a percentage of mitochondrial / nuclear genes lower than 20 (Fig. 5 below).

Fig. 5. Plot showing the percentage of mitochondrial genes in lung DTCs as a function of counts and features detected. Highlighted in red are the cells excluded from our analysis due to higher mitochondrial genes.

The ratio of mitochondrial / nuclear gene expression levels and the total number of detected features in those cells were analyzed. UMAP plots reveal that M DTCs are characterized by **lower** percentage of mitochondrial genes expression (< 5%, UMAP plot below in the center) than the more E DTCs (Fig. 6).

Fig. 6. UMAP showing the percentage of mitochondrial genes in lung DTCs.

We assessed expression levels of the stress-related genes *JUN*, *FOS*, *KCNQ1OT1*, and *EGR1* in the lung DTCs and found that clusters 0, 2 and 3 expressed *JUN*, *FOS*, and *EGR1*, while only clusters 0 and 2 expressed *KCNQ1OT1* (Fig. 7). Therefore, many DTCs- irrespective of their E or M state, express those genes.

Fig. 7. UMAP showing the expression of stress-related genes in lung DTCs.

In line with their decreased RNA features, we found RFP to be decreased in the M cells, but also differentially expressed among E cells (Fig.8).

Fig. 8. UMAP showing the RNA features in lung DTCs.

While we found that the overall transcriptional activity is decreased in the inactive cells (1,3), the expression of a subset of genes was increased (Fig.9 and attached excel sheet).

Fig 9. Heatmap showing the top differentially expressed genes in the 4 different clusters of lung DTCs.

Finally, most of the biological conclusions obtained by the scRNAseq have been validated by at least one molecular approach. Thus, we have shown by qPCR that the expression of *ZEB1*, *ZEB2*, and *VIM* was increased in MCF-7 and METS15 (PDX) lung DTCs compared to primary tumor cells (Figure 3 e-g + Supplementary Figure 4d,e). We also demonstrated by qPCR that many of the genes that characterized dormant MCF-7 DTCs, were also increased in lung DTCs from a PDX (METS15) (Supplementary Figure 9g). In addition, we show by IHC that ER protein levels were decreased in lung DTCs compared to their respective primary tumor cells (Supplementary Figure 9 d,e), in line with the hallmarks shown in the scRNAseq.

Reviewer #3 (Remarks to the Author):

The authors have done a masterful job of addressing my previous concerns, as well as those of the other reviewers.

We thank reviewer #3 for her/his appreciation of our revised manuscript and taking the time to look at the other reviewers' comments and our reply to them.

REVIEWER COMMENTS

Reviewer #1 (Remarks to the Author):

The authors addressed the comments very well. Especially the authors did a great job revising the conclusions to be consistent with the data. The revised manuscript is ready for publication.

Reviewer #2 (Remarks to the Author):

My comments below relate to previous questions remaining after various review rounds.

(1) Regarding naturally occurring E vs M cells: authors argue that M cells in their model constitute 0.6% of tumor cells in the primary site and that FACS sorting the required cells for transplantation would require more than 8 hours. It is unclear to me why authors do not consider using dedicated technologies for enrichment of rare cell populations (e.g. MACS columns or pull down with Ab-beads etc.) like it is done in many laboratories that deal with the same issues. RNA velocity does not contribute sufficiently to resolve this point. Based on this, I still consider the question unanswered.

(2) Regarding scRNA seq analysis: unfortunately, the provided QC data confirms a worrisome scenario, i.e. M-labeled cells are the ones with lowest number of features detected. Percent of mito genes is not really helpful in this case, given evident bias. Reason for asking to show RFP expression in the previous round was because I was hoping this to be the same (or very similar) across all cells, i.e. be a good control to exclude technical issues. Given that this is clearly not the case and M cells are showing significantly lower RFP expression, conclusions on gene expression or velocity that the authors infer from their data are not sufficiently sustained.

Reviewer #4 (Remarks to the Author):

The authors have proposed a mechanism for revert dormancy in ER+ cancer cells that depend on epithelial-mesenchymal plasticity. To my understanding, the dormancy mechanism is as follows: ER+ cells enter a non-proliferative state of dormancy by at least partially undergoing EMT, or as the authors put it, "exhibiting epithelial-mesenchymal plasticity" (EMP). The "awakening" of these dormant cells is associated to a return to the epithelial state guided by signals in the tumor microenvironment. To study this transition mechanism, the authors performed scRNA-seq of the disseminating tumor cells. The authors identified a small population of dormant cells that expressed mesenchymal genes and is this identified as a population of dormant cells exhibiting EMP. Finally, to demonstrate that quiescent, mesenchymal cells revert to an epithelial, proliferative state, the authors computed the RNA velocity field. Finally, they concluded from the directionality of RNA velocity arrows that transitions happen from the dormant mesenchymal state toward the epithelial state.

Overall, the authors have provided strong evidence for their proposed mechanism. However, I have some observations about the newly added RNA velocity analysis, which would greatly improve the manuscript. Some additional works are needed as detailed below:

-First, the RNA velocity map for the larger MCF7 dataset (figure S10A) is very difficult to interpret due to the large number of cells. I would suggest a different way to plot the RNA velocity arrows given the large number of cells in this dataset. The scVelo package has plotting feature that capture "average RNA velocity field" rather than the individual RNA velocity of single cells.

-Conversely, for the lung dataset exhibits some directionality (figure S10B), the authors based their conclusions on cells "at the bridge between inactive and active states", stating that these cells point toward the epithelial state. However, the concept of cells at the bridge is not defined at all. So, the authors should better define, and perhaps provide some visual help on the figure, to explain their reasoning.

-The panels C, D in supplementary figure 10 could provide further evidence on whether epithelial/mesenchymal genes are being induced or inhibited in dormant mesenchymal cells. Unfortunately, the description of these panels does not clarify (1) if the data comes from MCF7 or lung dataset and (2) if all cells or only a portion of cells are plotted. Therefore, at the moment, I cannot extract any conclusion from this data. Please reorganize these figures and clearly explain which genes are from MCF7 and which are from lung and discuss the findings in more details.

-Finally, the typical timescales for switching between dormant and active states are unclear to me. Typically, RNA velocity is applied to developmental systems where cells transition between cell states rather quickly (i.e., on a timescale of days), but I suspect that these cells can remain dormant for much longer. Therefore, the limit and applicability of RNA velocity to this problem should be discussed more clearly, and the timescales for switching between dormant and active states should be discussed.

We thank all four reviewers for their constructive feedback that helped us improve our manuscript. Their comments are marked in black below, our response is in red and in blue are citations of pertinent passages of the manuscript.

REVIEWER

COMMENTS

Reviewer #1 (Remarks to the Author):

We thank Reviewer #1 for her/his comments and advice, and for commending our manuscript for publication.

The authors addressed the comments very well. Especially the authors did a great job revising the conclusions to be consistent with the data. The revised manuscript is ready for publication.

Reviewer #2 (Remarks to the Author):

We thank Reviewer #2 for her/his scrutiny in analyzing scRNAseq data and RNA velocity, and for the helpful comments. We provide a detailed response below.

My comments below relate to previous questions remaining after various review rounds.

(1) Regarding naturally occurring E vs M cells: authors argue that M cells in their model constitute 0.6% of tumor cells in the primary site and that FACS sorting the required cells for transplantation would require more than 8 hours. It is unclear to me why authors do not consider using dedicated technologies for enrichment of rare cell populations (e.g. MACS columns or pull down with Ab-beads etc.) like it is done in many laboratories that deal with the same issues. RNA velocity does not contribute sufficiently to resolve this point. Based on this, I still consider the question unanswered.

We acknowledge the reviewer's concerns regarding naturally occurring E vs. M cells.

We considered using MACS columns to enrich for the rare M-cells in the primary tumor, however, we were unable to find a good antibody to mark the M-population. Using MACS columns to deplete the E-cells would still not produce 100% pure M-cells and therefore we would expect an outgrowth of the E-cells. Hence, we would still need to FACS sort the cells to purify our population of interest in the absence of an adequate antibody. In addition, this experiment would require an additional 6-8 months of additional mouse work for a question we believe we have already addressed with RT-qPCR, ex vivo culture of DTCs and scRNA-seq velocity.

As the primary concern of the reviewer is the transition between M and E states, we would like to re-emphasize the data shown in Supplementary Figure 7a-g, which demonstrate the transition of M-cells to E-cells. Please see excerpt below:

"To this aim, we dissociated mammary glands and lungs from *NSG-EGFP* mice engrafted with *MCF-7:RFP* cells to single cells, plated them in 2D and applied drug selection to avoid overgrowth of mouse cells. *RFP*⁺ *MCF-7* cells derived from primary tumors proliferated within a few days and were confluent by 1-2 weeks (Supplementary Fig. 7a). The lung-derived DTCs resumed proliferation after 2 months and formed epithelial islets (Supplementary Fig. 7b). The brain DTCs, which had lower *CDH1* transcript levels than the lung DTCs (Fig. 3f), took longer to emerge from quiescence and formed epithelial islets at 4 months (Supplementary Fig. 7c). *CDH1* transcript levels were ultimately restored, and EMT-TFs, *ZEB1*, *ZEB2* and *VIM* transcripts decreased after serial passages in culture (Supplementary Fig. 7d-g), consistent with the hypothesis that reacquisition of an epithelial state enables cell proliferation.." Page 11 lines 1135-1145

We have now highlighted future experiments (outside the scope of the current manuscript) that would confirm this data and discuss the possible limitations of the RNA velocity experiments in the discussion Page 17/18 lines 2410-2512.

(2) Regarding scRNA seq analysis: unfortunately, the provided QC data confirms a worrisome scenario, ie. M-labeled cells are the ones with lowest number of features detected. Percent of mito genes is not really helpful in this case, given evident bias. Reason for asking to show RFP expression in the previous round was because I was hoping this to be the same (or very similar) across all cells, i.e. be a good

control to exclude technical issues. Given that this is clearly not the case and M cells are showing significantly lower RFP expression, conclusions on gene expression or velocity that the authors infer from their data are not sufficiently sustained.

We thank the reviewer for challenging us to emphasize the reliability of the M-state from our scRNA-seq data. Although our analysis identifies RFP as being expressed to a lesser extent in the M-state, we have indeed confirmed that these cells are healthy cells as only cells that passed all quality control (QC) parameters as outlined by experts in the single-cell transcriptomics field were included in the analysis (194/266 cells). Upon combining E-states and M-states, we only see a slight upregulation expression of *FOS* in the M-state while all other stress markers, *JUN*, *EGR1* are either not differentially expressed or have decreased expression in the case of *KCNQ10T1*. This argues against M-state cells being of low quality.

In addition to this, we would like to highlight that the reduced RNA-synthesis has been characterized as a marker/hallmark of G0/dormant cells by multiple groups (M. Pallis et al, 2013 BMC Pharmacol Toxicol). It has been found that cells in a quiescent state have 30% reduced RNA content compared to their proliferative controls while the protein content remained unchanged. In line with the M-state being RFP+ at the protein level by flow cytometry while having reduced number RFP mRNA transcripts (Samuel Marguerat, et al., 2012, Cell). We have also found that although the total mRNA levels are depleted in the M-state, these cells harbor a diverse transcriptome with a number of genes being highly expressed and therefore a marker of the M-state. To ensure that these differentially expressed genes are not an artefact of scRNA-seq, we would like to refer to immunofluorescence staining (below) of both markers of the M-state (P27) (below a. and Supplementary Figure 3b) and E-state (Estrogen receptor) (below b.).

These data confirm the existence of both the E and M state within the lung DTCs at the mRNA and protein level. We would also like to highlight that we have additionally validated the scRNA-seq data via qPCR of M-associated genes (please see Figure 3 e-g and Supplementary Figure 4 d,e).

Reviewer #4 (Remarks to the Author):

We thank the fourth reviewer for her/his constructive criticism and expertise in scRNAseq and RNA velocity. The feedback provided has helped us improve the data pertinent to RNA velocity and strengthen our conclusions.

The authors have proposed a mechanism for revert dormancy in ER+ cancer cells that depend on epithelial-mesenchymal plasticity. To my understanding, the dormancy mechanism is as follows: ER+ cells enter a non-proliferative state of dormancy by at least partially undergoing EMT, or as the authors put it, “exhibiting epithelial-mesenchymal plasticity” (EMP). The “awakening” of these dormant cells is associated to a return to the epithelial state guided by signals in the tumor microenvironment. To study this transition mechanism, the authors performed scRNA-seq of the disseminating tumor cells. The authors identified a small population of dormant cells that expressed mesenchymal genes and is this identified as a population of dormant cells exhibiting EMP. Finally, to demonstrate that quiescent, mesenchymal cells revert to an epithelial, proliferative state, the authors computed the RNA velocity field. Finally, they concluded from the directionality of RNA velocity arrows that transitions happen from the dormant mesenchymal state toward the epithelial state.

Overall, the authors have provided strong evidence for their proposed mechanism. However, I have some observations about the newly added RNA velocity analysis, which would greatly improve the manuscript. Some additional works are needed as detailed below:

-First, the RNA velocity map for the larger MCF7 dataset (figure S10A) is very difficult to interpret due to the large number of cells. I would suggest a different way to plot the RNA velocity arrows given the large number of cells in this dataset. The scVelo package has plotting features that capture “average RNA velocity field” rather than the individual RNA velocity of single cells.

As advised by the reviewer, we have now used the average RNA velocity field feature using the scVelo package shown in the new Figure S10A:

RNA smn res.0.5

We have amended the figure legend accordingly: “a. UMAP showing the steady-state RNA velocity arrows in MCF-7 cells at the primary site using the average RNA velocity field.”

-Conversely, for the lung dataset exhibits some directionality (figure S10B), the authors based their conclusions on cells “at the bridge between inactive and active states”, stating that these cells point toward the epithelial state. However, the concept of cells at the bridge is not defined at all. So, the authors should better define, and perhaps provide some visual help on the figure, to explain their reasoning.

We thank the reviewer for highlighting that our terminology of the ‘bridge’ was unclear. We have now highlighted the ‘bridge’ between the mesenchymal and epithelial cells and termed it ‘EMP bridge’. Please see New Supplementary Figure 10 b and changes in the text. Page 15, lines 1942-1954.

-The panels C, D in supplementary figure 10 could provide further evidence on whether

epithelial/mesenchymal genes are being induced or inhibited in dormant mesenchymal cells. Unfortunately, the description of these panels does not clarify (1) if the data comes from MCF7 or lung dataset and (2) if all cells or only a portion of cells are plotted. Therefore, at the moment, I cannot extract any conclusion from this data. Please reorganize these figures and clearly explain which genes are from MCF7 and which are from lung and discuss the findings in more details.

We apologize for the lack of clarity in this figure. We have now included further descriptions within the figures. Please see new Supplementary Figure 10 c, d. We have now added labels “Lung DTCs” next to panel c and d to guide the reader to which dataset is being analyzed.

-Finally, the typical timescales for switching between dormant and active states are unclear to me. Typically, RNA velocity is applied to developmental systems where cells transition between cell states rather quickly (i.e., on a timescale of days), but I suspect that these cells can remain dormant for much longer. Therefore, the limit and applicability of RNA velocity to this problem should be discussed more clearly, and the timescales for switching between dormant and active states should be discussed.

We thank the reviewer for bringing up this important point. We have now included the limitations of RNA velocity and the time scale in the discussion. Page 18, lines 2298ff:

“...This approach is intended for predicting cell states within a number of hours, likely a much shorter time-scale than the in vivo plasticity observed in patients. Yet, we are studying these DTCs at extremely late stages, where a number of cells are already transitioning from M to E. In addition to this, we cannot exclude that the majority of mesenchymal DTCs may have been lost during tissue dissociation. In order to distinguish between the two scenarios further studies would be required for example repeating these experiments using a lineage tracing barcoded library in combination with scRNA-seq.”

REVIEWERS' COMMENTS

Reviewer #4 (Remarks to the Author):

Happy with the revision on the RNA velocity study. I've no more comments.